# Predicting in-hospital outcomes of patients with acute kidney injury

Changwei Wu[1,22], Yun Zhang[2,22], Sheng Nie[3,22], Daqing Hong[1], Jiajing Zhu[2], Zhi Chen[2], Bicheng Liu [4], Huafeng Liu[5], Qiongqiong Yang[6], Hua Li[7], Gang Xu[8], Jianping Weng[9], Yaozhong Kong[10], Qijun Wan[11], Yan Zha[12], Chunbo Chen [13], Hong Xu[14], Ying Hu[15], Yongjun Shi[16], Yilun Zhou[17], Guobin Su[18], Ying Tang [19], Mengchun Gong [20,21], Li Wang[1], Fanfan Hou [3] ✉, Yongguo Liu [2] ✉ & Guisen Li [1] ✉

Acute kidney injury (AKI) is prevalent and a leading cause of in-hospital death worldwide. Early prediction of AKI-related clinical events and timely intervention for high-risk patients could improve outcomes. We develop a deep learning model based on a nationwide multicenter cooperative network across China that includes 7,084,339 hospitalized patients, to dynamically predict the risk of in-hospital death (primary outcome) and dialysis (secondary outcome) for patients who developed AKI during hospitalization. A total of 137,084 eligible patients with AKI constitute the analysis set. In the derivation cohort, the area under the receiver operator curve (AUROC) for 24-h, 48-h, 72-h, and 7-day death are 95·05%, 94·23%, 93·53%, and 93·09%, respectively. For dialysis outcome, the AUROC of each time span are 88·32%, 83·31%, 83·20%, and 77·99%, respectively. The predictive performance is consistent in both internal and external validation cohorts. The model can predict important outcomes of patients with AKI, which could be helpful for the early management of AKI.

Acute kidney injury (AKI) is a leading cause of in-hospital death worldwide, with a prevalence of about one-fifth in hospitalized patients[1–4]. A previous meta-analysis of more than 77 million hospitalized patients from 952 studies showed that the pooled incidence of AKI was 21%, and the in-hospital mortality rate of AKI patients was approximately 21%[1]. Among them, patients with AKI stage 3 and those receiving renal replacement treatment had a mortality rate of 42% and 46%, respectively[1]. Two large-scale inpatient studies in China also showed that the prevalence of AKI was 2·3% and 11·6%, respectively[5,6]. Because of the high prevalence and mortality of in-hospital AKI, early recognition and treatment are critical to successful outcomes[1].

The International Society of Nephrology launched the 0by25 Initiative that aims to eliminate preventable deaths from AKI by 2025[1]. Meanwhile, despite extensive efforts in recent years, no efficient treatments can obviously increase or accelerate kidney recovery[4]. The American Society of Nephrology has launched a new initiative (AKI! Now) to promote excellence in the prevention and treatment of AKI by

building a foundational program, to reduce morbidity and associated mortality and to improve long-term outcomes[7].

Following AKI, morbidity and mortality can be decreased through early detection and intervention. Our previous study revealed that in-hospital AKI is caused by heterogeneous causes in various departments, resulting in about three-quarters of patients missing diagnosis[5]. It increased the risk of clinical adverse events in those overlooked AKI patients. Artificial intelligence (AI) could be useful for time-sensitive applications in recognizing, alerting, and providing treatment suggestions for AKI[8]. Machine learning-based models could detect AKI early and provide clinicians with much earlier intervention opportunities[9–14]. Previous predictions were conducted in special settings, including in patients with hospital-acquired AKI[15], postoperative AKI[13], AKI secondary to cancer[8], critical illness in the intensive care unit (ICU)[16,17], and patients admitted to the emergency department[18]. Few prediction models can cover general patients as well as critically ill patients or those undergoing major surgery in general hospitals. Due

to the high heterogeneity of patients, it is difficult to independently validate these prediction models in general hospitals.

However, the patient's status is constantly changing, so the prediction of a machine-learning model should be automatically adjusted based on emerging clinical information and new changes. Tomašev successfully developed a deep learning model for the continuous risk prediction of AKI in hospitals based on a large dataset[9]. Continuous and dynamic prediction can offer prompt opportunities for identifying patients at risk within a window that enables early treatment. We have previously reported a machine-learning model for AKI using a recurrent neural network (RNN) algorithm based on parameters within three days prior to hospitalization; it effectively predicted AKI and performed significantly better with time features than without it[19].

Although there are many models that can identify patients at risk for AKI, however, once patients develop AKI, few models predict the risk of clinically important outcomes (such as hospital death or dialysis) in AKI patients[20]. Predictive models for clinically important outcomes could be helpful in guiding the early management of AKI patients.

In this work, based on a nationwide multicenter cooperative network across China, we present a deep learning model to dynamically predict the risk of in-hospital death and dialysis for patients who developed AKI during hospitalization.

## Results

### Participants
Of the 7,084,339 patients in Chinese Renal Disease Data System (CRDS), 5,514,410 with ≤1 serum creatinine (SCr) measurement during hospitalization were excluded. Overall, 200,881 patients developed AKI during hospitalization. After applying the exclusion and inclusion criteria, the final dataset consisted of 137,084 patients (Fig. 1).

### Characteristics of the cohorts
Table S1 showed the baseline characteristics of the derivation, internal, and external validation cohorts. The external validation cohort had a higher percentage of patients with AKI stage 0, higher incidence of mortality, need for dialysis, use of mechanical ventilation, hypertension, and diabetes, higher Charlson comorbidity scores, and a lower percentage of patients who underwent major surgery and ICU admission than the other cohorts. For laboratory parameters, SCr, C-reactive

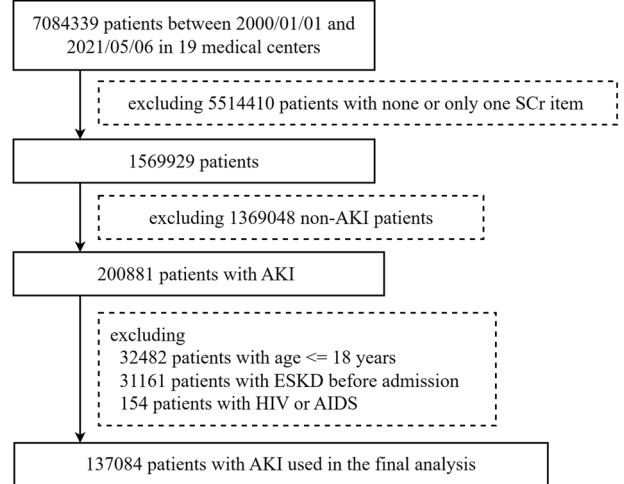

**Fig. 1 | Flow chart of the study population selection.** We selected patients who developed AKI during hospitalization for further screening. The exclusion criteria were as follows: (1) patients who had less than two SCr results during hospitalization; (2) patients <18 years old; (3) patients who had HIV or AIDS; and (4) patients who had end-stage kidney disease (ESKD, defined as maintenance dialysis, kidney transplantation, or eGFR <15 ml/min per 1.73m2). SCr, serum creatinine. HIV human immunodeficiency virus, AIDS acquired immunodeficiency syndrome.

protein, chloride, procalcitonin, and erythrocyte sedimentation rate were higher in the external validation group.

We compared the baseline characteristics between the death and survival cohorts (Table S2). Overall, 1864 patients with AKI (1.38%) died in hospital. These dead patients had higher baseline SCr and proteinuria, as well as more severe AKI stage. The prevalence of hypertension and diabetes in the death group was higher than that in the survival group, and Charlson's complication score was also higher than that in the survival group. In the death group, more patients received dialysis or mechanical ventilation, and stayed in the ICU (Table S2).

### Performance of the prediction model for death
In the derivation cohort, the AUROCs for predicting 24 h, 28 h, 72 h, and 7d mortality were 95.05%, 94.23%, 93.53%, and 93.09%, respectively. The internal validation cohort's AUROCs were 93.58%, 92.45%, 93.02%, and 87.03% at 24 h, 48 h, 72 h, and 7d. In the external validation cohort, the AUROCs at 24 h, 48 h, 72 h, and 7d were 92.43%, 92.16%, 88.36%, and 88.32%, respectively. We presented the accuracy, F-score, precision, and recall in Fig. 2.

### Performance of the prediction model for dialysis
Figure 3a showed the AUROCs for predicting dialysis. In derivation cohorts, the AUROCs for 24 h, 28 h, 72 h, and 7d were 88.32%, 83.31%, 83.20%, and 77.99%, respectively. The internal validation cohort's AUROCs were 88.33%, 82.73%, 83.09%, and 77.25% at 24 h, 48 h, 72 h, and 7d, respectively. In the external validation cohort, the AUROCs for dialysis were 74.18%, 77.58%, 75.21%, and 69.34% at 24 h, 48 h, 72 h, and 7d, respectively. Other evaluation indicators were presented in Fig. 3.

### Comparing with baseline algorithms
The results of baseline comparison were provided in Supplementary Table S3. Whether predicting death or dialysis, BiLSTM (original) and BiSingleLSTM (appending single-direction LSTM) both had relatively poor predictive capacity.

### Subgroup analysis of the prediction model
We also conducted a subgroup analysis of previously reported risk factors related to the mortality or dialysis, including, age, gender, hypertension, diabetes, AKI stage, baseline SCr, length of ICU stay, and major surgery. Figure 4 had shown the AUROC curve at 24-h, 28-h, 72-h, and 7-day for predicting death and dialysis. All AUROCs of various subgroups were almost more than 80% in predicting death and more than 75% in predicting dialysis, which indicated that AKI event prediction model (AKIEPM) performed well in the above-mentioned clinical situations.

## Discussion
The early prediction of disease progression could improve healthcare. An estimated 11% of in-hospital deaths are due to the failure to promptly recognize and treat deteriorating patients[9]. Improving the prediction ability of deep learning models remains a challenge in AKI prediction[8,9,21–23]. AI-driven health interventions have been successfully used to diagnose and assess patients' morbidity or mortality risk, predict and surveillance disease outbreaks, and plan health policies[21,24]. Contemporary AI applications can accurately predict the onset of AKI before notable biochemical changes occur and the need for future dialysis to some extent[8,9,19,21–23,25]. Additionally, a clinical decision support system can result in a small but sustained decrease in-hospital mortality, dialysis use, and length of hospitalization[26]. In this study, we developed AKIEPM based on deep learning to predict in-hospital outcomes of patients with AKI. AKIEPM had good predictive performance for in-hospital death and the need of dialysis, which may prompt preventive action to reduce the risk of unfavorable outcomes.

AKIEPM predicted the outcome of patients with AKI at 24 h, 48 h, 72 h, and 7d by extracting latent features from patient data at

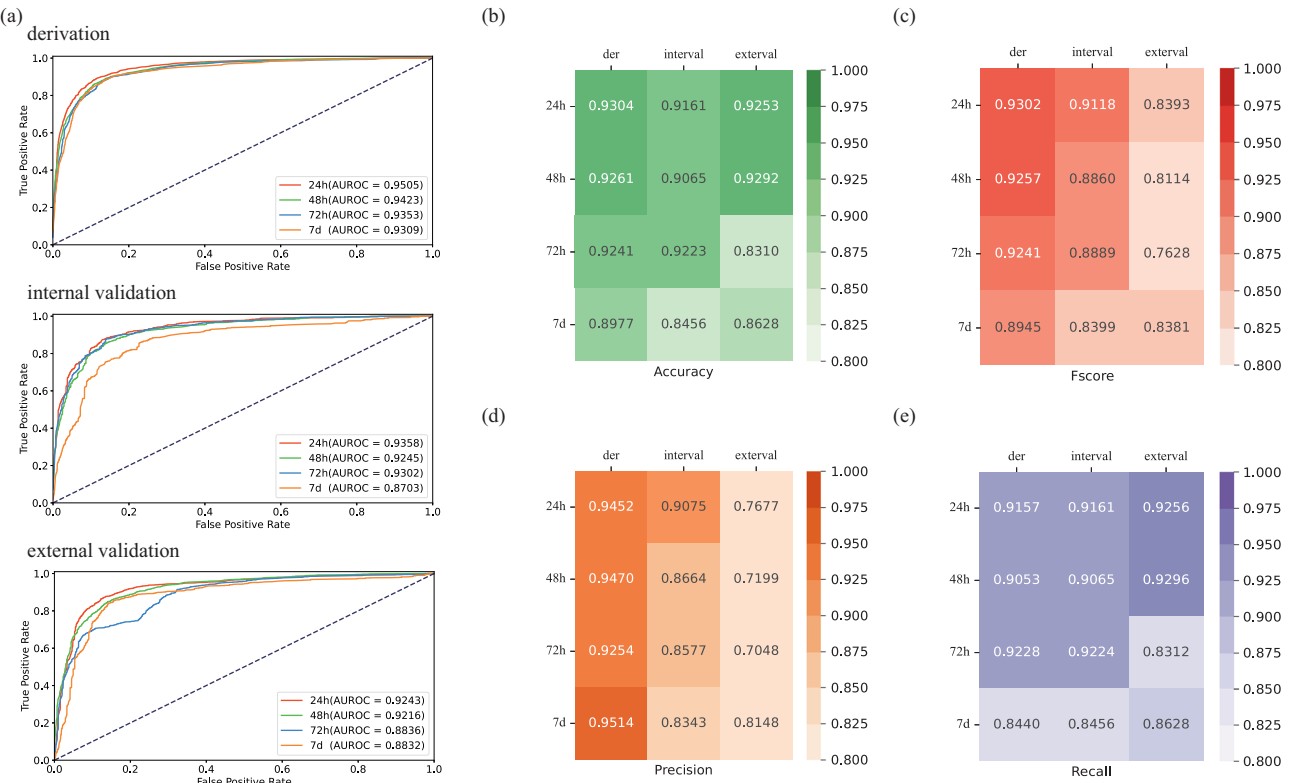

**Fig. 2 | Various evaluation indicators of the prediction model of death.** It showed the AUROC curves (**a**), accuracy (**b**), F-score (**c**), precision (**d**), and recall (**e**) for predicting 24-h, 28-h, 72-h, and 7-day mortality in derivation, internal, and external validation cohorts. In this study, we train the deep learning model in derivation cohort and test in internal validation cohort with 100 epochs and validate in external validation cohort to obtain the results.

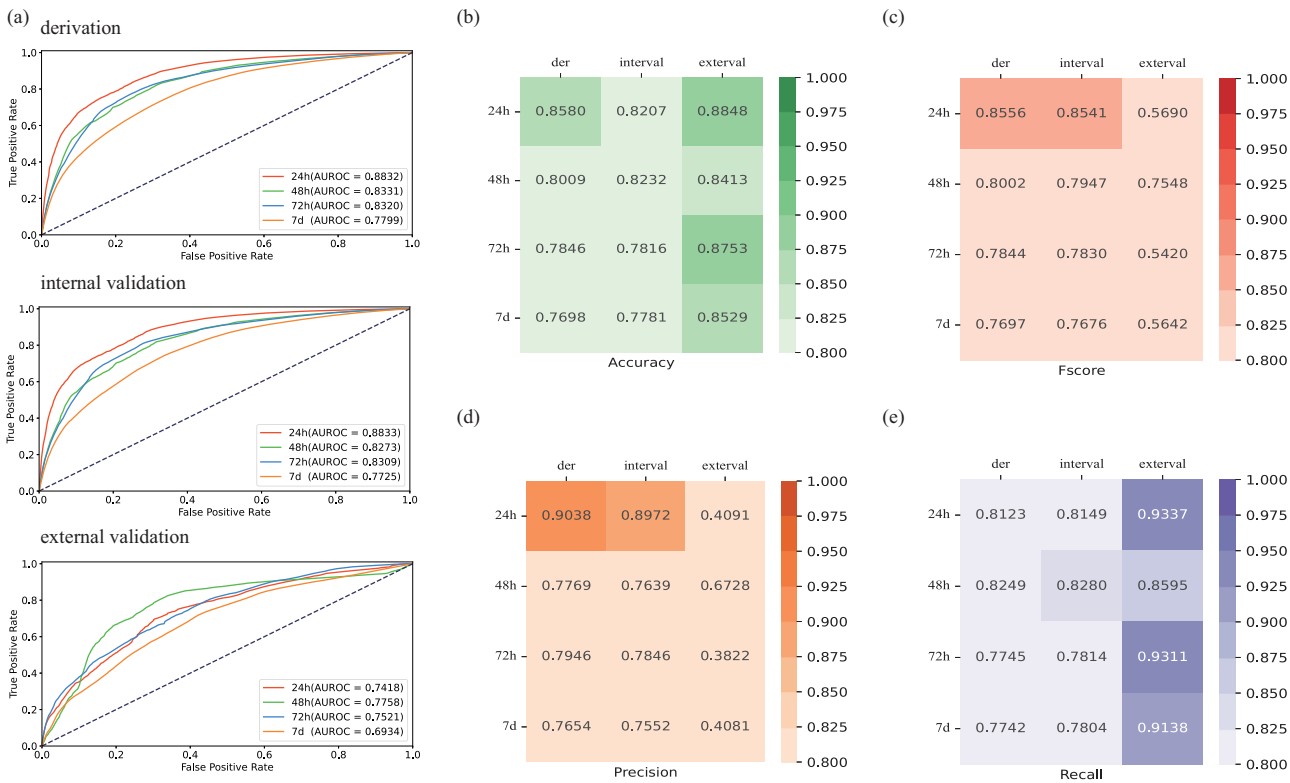

**Fig. 3 | Various evaluation indicators of the prediction model of dialysis.** It showed the AUROC curves (**a**), accuracy (**b**), F-score (**c**), precision (**d**), and recall (**e**) for predicting 24-h, 28-h, 72-h, and 7-day dialysis in derivation, internal, and external validation cohorts. In this study, we train the deep learning model in derivation cohort and test in internal validation cohort with 100 epochs and validate in external validation cohort to obtain the results.

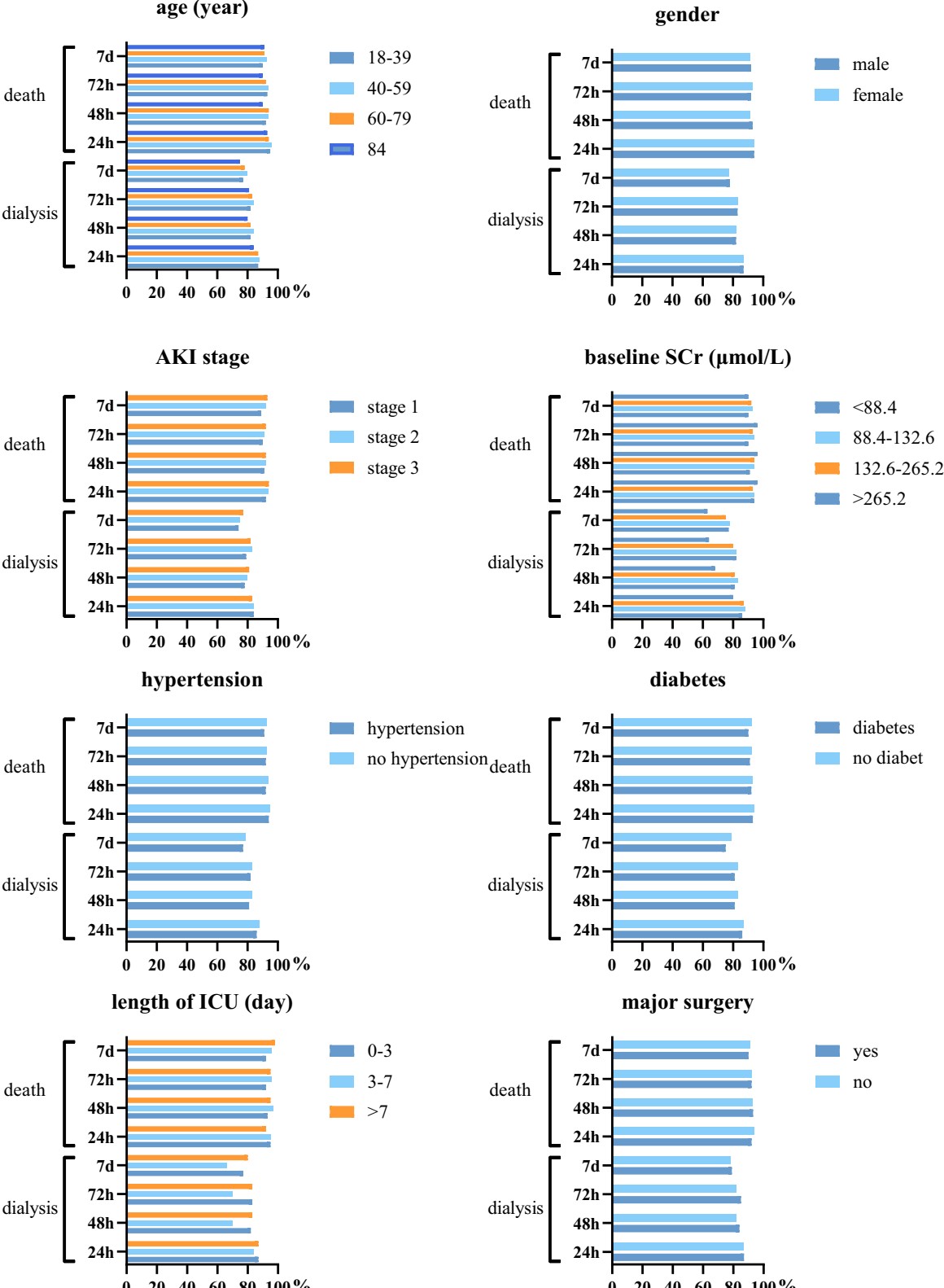

**Fig. 4 | Prediction of death and dialysis in Subgroup cohorts.** We conducted subgroup analyze of age, gender, hypertension, diabetes, AKI stage, baseline SCr, length of ICU stay, and major surgery. *X*-axis was the value of AUROC, and *Y*-axis was model predicting death and dialysis at 24-h, 28-h, 72-h, and 7-day.

various times and learning the association between features. Through context embedding, which can make greater use of each variable's semantic properties, the model took the values of each variable into account. Bidirectional long- and short-term memory units were simultaneously adopted to learn bidirectional temporal correlations from patient data. It meant that the outcome of the patient may be predicted by using the forward and reverse correlation of data collected in the past and in the present. Our continuous prediction model was developed from a large, retrospective, and longitudinal dataset that covers diverse clinical

environments in general hospitals. Experimental results demonstrated the effectiveness of our predictive model.

To further verify the performance of the model, we used BiLSTM and BiSingleLSTM[27] as a comparison. We considered that BiLSTM and BiSingleLSTM gained poor performance might be because they do not encode the clinical variables or they were stuck in local optimality and stopped early.

For predictive alerts to be effective, they must empower clinicians to act before a major clinical event occurs[28]. Therefore, we used dynamic prediction methods to predict the future in-hospital outcomes at each time point. In addition, to achieve generalization, we developed and validated AKIEPM in different hospital patient cohorts. Results demonstrated that the transportability of our prediction model between different hospitals was good.

We projected four time periods in this study (24 h, 48 h, 72 h, and 7d). The results revealed that as the time breadth was increased, the prediction performance gradually decreased, which was a projected outcome. Furthermore, the model's performance in predicting the need of dialysis was lower than that of death. One possible reason was that different physicians make different decisions about whether and when to initiate dialysis for a patient based on factors, including disease status, personal willingness, and accessibility to dialysis.

The advantages of AKIEPM included its novelty for predicting the occurrence of death or dialysis by creating a bidirectional LSTM unit. Additionally, AKIEPM could contain all data before the time point for prediction, which was useful for dynamically assessing changes in a patient's condition. Moreover, we projected four time periods, which provided a window of up to 72 h for clinical intervention. Finally, AKIEPM could predict the occurrence of outcomes, allowing patients to be stratified for targeted treatment and management to reduce mortality.

There were several limitations in our study. First, AKI is a clinically complex condition with various causes, which may influence the disease course and outcomes of AKI; however, we did not conduct a thorough subgroup analysis and model establishment for the different AKI subgroups. Second, AKIEPM had difficulty determining which factors were responsible for the increased risk of in-hospital death; hence, it was difficult to suggest targeted treatment. Third, AKIEPM was conducted on previous data, and whether it could help improve prognosis must be confirmed in prospective controlled trial[29]. Despite the performance of our model when compared to those in the literature, future studies should evaluate and independently validate our model to establish its clinical utility and effects on decreasing unfavorable in-hospital and outpatient outcomes as well as to explore the role of AKIEPM in researching strategies for delivering preventive care for patients with AKI. AKIEPM could potentially become a crucial part of routine clinical pathways for AKI management.

In summary, we introduced a deep learning approach for the dynamic prediction of in-hospital death or need of dialysis in patients with AKI. The model was validated using a large number of patients with AKI as an internal cohort and an external cohort. Those who were at risk for AKI could be identified through AKIEPM.

## Methods

The conceptual framework for our developing dynamic prediction model was presented in Fig. 5.

### Data description

The study protocol was approved by the Medical Ethics Committee of Sichuan Provincial People's Hospital (approval number: 2022-83), and Nanfang Hospital, Southern Medical University (approval number: NFEC-2019-213), which waived the requirement for patient informed consent due to the retrospective nature of the study. This study was also approved by the China Office of Human Genetic Resources for Data Preservation Application (approval number: 2021-BC0037) and was performed by the Strengthening the Reporting of Observational Studies in Epidemiology (STROBE) guidelines.

The study population was derived from CRDS, a large multicenter retrospective study cohort of 7,084,339 patients hospitalized at 19 medical centers throughout China from 1 January 2000 to 26 May 2021. The dataset consisted of information from hospital electronic health records in digital format. The number of independent entries in the dataset was ~2.8 billion, including 37,224 features. We extracted adult patients' data, including outpatient visits, admissions, diagnoses as International Statistical Classification of Diseases and Related Health Problems codes, surgical procedures (including date, names, and codes [ICD-9-CM-3]), vital signs, stay-in ICU, mechanical ventilation, laboratory results (including—but not limited to—biochemistry, hematology, cytology, microbiology, and histopathology), medications and prescriptions, orders, dialysis (including hemodialysis, peritoneal dialysis, continuous renal replacement therapy), and in-hospital death.

Patients who developed AKI during hospitalization (including community-acquired AKI) were selected for further screening. The exclusion criteria were as follows: (a) patients who had less than two SCr results during hospitalization; (b) patients <18 years old; (c) patients who had human immunodeficiency virus or immunodeficiency syndrome; and (d) patients who had end-stage kidney disease (ESKD, defined as maintenance dialysis, kidney transplantation, or eGFR <15 ml/min per 1.73 m$^2$). Patients who had undergone dialysis prior to developing AKI were also excluded when dialysis was analyzed as an in-hospital outcome. The flow chart of study enrollment is shown in Fig. 1.

### Definitions

AKI was defined according to the Kidney Disease Improving Global Outcomes (KDIGO) clinical practice guideline[3]. Due to the lack and

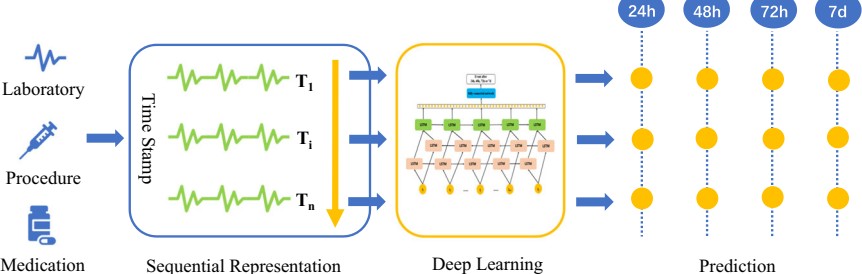

**Fig. 5 | Conceptual model of continuously predicting AKI in-hospital outcomes.** First, we collect the patient information, such as laboratory, procedure, medication, etc. Second, we construct a sequential representation of electronic health records by merging patient data in 24 h. Third, we propose AKIEPM based on deep learning. Fourth, we predict the occurrence of death or need for dialysis at 24 h, 48 h, 72 h, and 7d.

accuracy of urine output, we employed SCr to define AKI as an increase by 0.3 mg/dL within 48 h or a 50% increase from baseline within 7 days. The definition of AKI was based on the dynamic criteria described by Nanfang Hospital[6]. The SCr data collected during hospitalization were sorted in increasing order by sequential test time. For any time point $t$, a baseline average SCr was dynamically defined as the average value of SCr within 7 days before the point $t$, then each available SCr value within 7 days after point $t$ was compared with this baseline average Scr. The date of AKI diagnosis was defined as the earliest day on which the SCr change met the KDIGO criteria. The baseline value for further analysis in patients with AKI was defined as the value at the time of AKI diagnosis. AKI stages were determined by the peak SCr level after AKI detection, with a rise of less than 100% indicating stage 1, a rise of 100% or more indicating stage 2, and a rise of 200% or more over baseline indicating stage 3.

The primary and secondary outcomes were in-hospital death and the need for dialysis, respectively. In this study, dialysis included temporary or maintenance hemodialysis and peritoneal dialysis, and continuous renal replacement therapy (CRRT).

Diagnosis codes for admission and discharge were used to identify comorbidities. The Charlson comorbidity score was used to calculate the burden of comorbidity[30]. The Chinese surgical operation grading system was used to classify the procedure from grade 1 to grade 4 based on its difficulty, complexity, and risk. In this study, major surgery was defined as grade 4 surgery, as well as grade 3 invasively surgical operation in major body cavities (heart, intracranial, chest, abdomen, or pelvic).

## Construction of derivation and validation cohorts

To train and validate the performance of the prediction model, we divided the patients into the derivation, internal validation, and external validation cohorts with a 7:2:1 ratio. For the external validation cohort, we adopted a traversal strategy to enumerate different combinations of hospitals and chose one hospital-combination (including three hospitals) wherein the number of patients was the closest to 10% of the overall cohort (14,610 patients). From patients in the remaining hospitals except for the three hospitals, 27,217 patients (20% of the total) were randomly selected as the internal validation cohort, then all other patients, ~70% of the total, were selected as the derivation cohort (Fig. 6). For classification imbalance problems (death/no death; dialysis/no dialysis), we adopt a random oversampling strategy to decrease the imbalance problem.

## Construction of sequential representation

We first constructed a sequential representation of electronic health records by merging patient data in 24 h. Although the onset of symptoms, laboratory results, operations, prescriptions, medications, diagnoses, and events may be different, they may happen within a day. If we constructed sequential data at each time point, this will result in several missing data. For example, if the operation time was 14:02 whereas others were different, the record with this operation will miss other data at this time. To decrease the amount of missing data, we merged the data in 24 h to construct a sequential representation for dynamically predicting death or dialysis within the next 24 h, 48 h, 72 h, and 7 days (d).

$\mathbf{P} = \{p_1, \ldots, p_i, \ldots, p_N\}$ denoted the set of patients, where $p_i$ was the $i$-th patient and $N$ was the number of patients. We used $\mathbf{t}^p = \{\mathbf{t}_1^p, \ldots, \mathbf{t}_t^p, \ldots, \mathbf{t}_T^p\}, \mathbf{t}_t^p \in \mathbb{R}^V$ to represent all variables at time $t$ of patient $p_i \in P$ (in this paper, we adopted the patient data after AKI diagnosis), where $T$ denoted the number of days in admission of patients and $V$ denoted the number of variables. We set binary label $y^p \in \{0, 1\}$ to represent the outcome of death or dialysis prediction in 24 h, 48 h, 72 h, and 7d. The merge rules were as follows, wherein we took the detection time of creatinine as the standard time (Fig. 7):

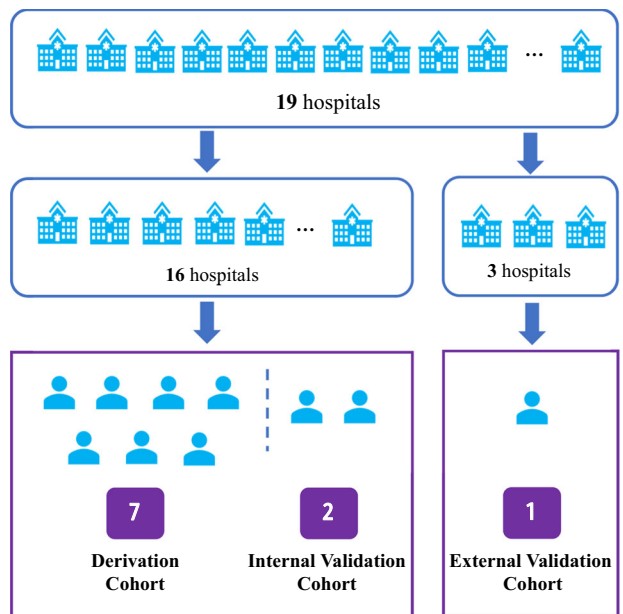

**Fig. 6 | Flow chart of the construction of derivation, internal validation, and external validation cohorts.** To train and validate the performance of AKIEPM, we divided the patients into the derivation, internal validation, and external validation cohorts with a 7:2:1 ratio. For the external validation cohort, we chose three hospitals wherein the number of patients was the closest to 10% of the overall cohort (14,610 patients). From patients in the remaining hospitals, 27,217 patients (20% of the total) were randomly selected as the internal validation cohort, then all other patients (~70% of the total) were selected as the derivation cohort.

(1) If there was no SCr result within 24 h, all laboratory values were combined with that record at the latest SCr measurement. If there were multiple values for the same indicator, the latest value was taken.

(2) If there was one creatinine value within 24 h, other laboratory values were combined with that record at the time of SCr measurement. If there were multiple values for the same indicator, those simultaneous with the SCr measurement or the closest were taken.

a. If there were multiple laboratory values, the maximum temperature, lowest (<50) and highest (>50) pulse rates, minimum blood pressure, maximum respiration rate, and laboratory values closest to the time of SCr measurement were taken.

b. If there were multiple values for the operation, medications, prescriptions, diagnosis, and events per day, we obtained those closest to when the SCr levels were measured.

(3) and (4) If there were two SCr values within 24 h, laboratory values before and after the first SCr measurement were combined with the first and second SCr measurements, respectively. If there were multiple values for other laboratory indicators, refer to (2).

## Prediction model

We proposed a novel AKI event prediction model with bidirectional LSTM unit[31] for predicting the occurrence of death or need for dialysis at 24 h, 48 h, 72 h, and 7d. Table S4 showed the parameters included in AKIEPM. AKIEPM includes three steps: embedding, feature extraction, and output (Figure S1). We first developed 8 models to predict 24 h, 48 h, 72 h, and 7d outcomes of death/no death, dialysis/no dialysis, and then we integrated them into AKIEPM.

(1) Embedding. Capture the dependence of different variables by encoding their values into a continuous vector[32].

(2) Feature extraction. RNN was widely used to deal with sequential data problems; however, its prediction performance dropped when the length of the sequence increases by 24 h[33]. An LSTM unit was adopted to construct a bidirectional LSTM layer to capture the relevance among the sequential representation of

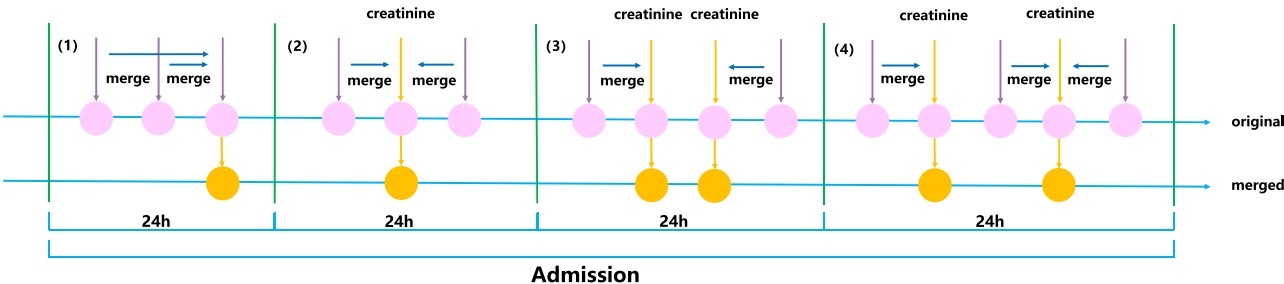

**Fig. 7 | Schematic of the merge rules in the AKI event prediction model AKIEPM.** It showed the situations of no SCr result within 24 h (1), one SCr result within 24 h (2), and two SCr results within 24 h (3), and (4) for constructing a sequential representation of electronic health records by merging patient data in 24 h.

patients before and after the time, energizing the dynamic prediction every 24 h. Thus, the bidirectional LSTM layer extracted the features from the forward and reverse inputs. The next one-way LSTM layer fused the bidirectional output and obtained the hidden representation of all variables.

(3) Output. The hidden representation was fed to the *softmax* layer, which predicted death or dialysis within 24 h, 48 h, 72 h, and 7d.

### Evaluation
We adopted precision, recall, F-score, and AUROC to evaluate the performance of the proposed model. To evaluate the effectiveness of our model, we compared it with algorithms BiLSTM and BiSingleLSTM that had similar deep learning architecture. Compared with baselines, AKIEPM encoded the values of patient data into a continuous vector, which can capture the meaning of different indexes to a certain extent. Meanwhile, feature extraction can capture the relevance among the sequential representation before and after the time to extract valuable hidden features concerned with the outcomes of patients with AKI. The time of external verification cohort concentrates on 2016–2020, which is close to the current time, with respect to derivation and internal validation cohorts, we can consider that the performance in ultimate deployment can be estimated by the performance in external verification cohort to a certain extent.

### Statistical analyses
The Chi-square test was used to compare categorical variables that were presented as frequencies and percentages. The *t*-test was used to compare normally distributed variables that were reported as mean and standard deviation. Non-normally distributed variables were expressed as median and interquartile range and compared using the Kruskal–Wallis H test. *P*-values were two-tailed, and $p < 0.05$ was considered significant. R software v3·1·1 was used for all analyses.

### Reporting summary
Further information on research design is available in the Nature Portfolio Reporting Summary linked to this article.

## Data availability
Data for figures are provided with this paper and are available at https://github.com/yunzhangwww/AKIEPM. Currently, CRDS data is only free to share with participating collaborators who have signed the cooperation agreement. If you are willing to cooperate, please contact the CRDS database administrator via the official email address: ncrckd@163.com (http://www.crds-network.org.cn/#/joinUs/joinus), and they will respond within 1 month.

## Code availability
The data analysis by AKIEPM was coded by TensorFlow 1.15 and Python 3.7 with patient data, whose code is available at https://github.com/yunzhangwww/AKIEPM.

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

## Acknowledgements

This work was supported by the National Natural Science Foundation of China (82070690, U19A2004, LW) and Medico-Engineering Cooperation on Applied Medicine Research Center, University of Electronic Science and Technology of China (ZYGX2021YGLH012, G.L. and Y.L.). We are also grateful for the support of Digital Health China Technologies Co. We would like to thank Editage (www.editage.cn) for English language editing.

## Author contributions

G.L., Y.L., L.W., and F.H. initiated the project and the collaboration. G.L., Y.L., C.W., Y.Zhang, and S.N. designed the overall study. Y.L., Y.Zhang, J.Z., and Z.C. designed and implemented the model. Y.Zhang and C.W. extracted the study cohort, cleaned up the data, and performed all experiments. G.L. and C.W. contributed their clinical expertise. F.H. and S.N. proposed and created the CRDS. F.H., S.N., D.H., B.L., H.Liu, Q.Y., H.Li, G.X., J.W., Y.K., Q.W., Y.Zha, C.C., H.X., Y.H., Y.S., Y.Zhou, G.S., Y.T., M.G., and L.W. provided clinical data for CRDS. C.W. and Y.Zhang wrote the paper, with revision advice provided by G.L., Y.L., L.W., and F.H. All authors have read and approved the manuscript.

## Competing interests

The authors declare no competing interests.

## Additional information

¹Department of Nephrology and Nephrology Institute, Sichuan Provincial People's Hospital, School of Medicine, University of Electronic Science and Technology of China, 610072 Chengdu, China. ²Knowledge and Data Engineering Laboratory of Chinese Medicine, School of Information and Software Engineering, University of Electronic Science and Technology of China, 610054 Chengdu, China. ³National Clinical Research Center for Kidney Disease, State Laboratory of Organ Failure Research, Division of Nephrology, Nanfang Hospital, Southern Medical University, 510515 Guangzhou, China. ⁴Institute of Nephrology, Zhongda Hospital, Southeast University School of Medicine, 210000 Nanjing, China. ⁵Key Laboratory of Prevention and Management of Chronic Kidney Disease of Zhanjiang City, Institute of Nephrology, Affiliated Hospital of Guangdong Medical University, 524000 Zhanjiang, China. ⁶Department of Nephrology, Sun Yat-Sen Memorial Hospital, Sun Yat-Sen University, 510515 Guangzhou, China. ⁷Sir Run Run Shaw Hospital, Zhejiang University School of Medicine, 310000 Hangzhou, China. ⁸Division of Nephrology, Tongji Hospital, Tongji Medical College, Huazhong University of Science and Technology, 430000 Wuhan, China. ⁹Department of Endocrinology, The First Affiliated Hospital of USTC, Division of Life Sciences and Medicine, University of Science and Technology of China, 230000 Hefei, China. ¹⁰Department of Nephrology, the First People's Hospital of Foshan, 528000 Foshan, China. ¹¹The Second People's

Hospital of Shenzhen, Shenzhen University, 518000 Shenzhen, China. [12]Guizhou Provincial People's Hospital, Guizhou University, 550000 Guiyang, China. [13]Department of Critical Care Medicine, Maoming People's Hospital, 525000 Maoming, China. [14]Children's Hospital of Fudan University, 200000 Shanghai, China. [15]The Second Affiliated Hospital of Zhejiang University School of Medicine, 310000 Hangzhou, China. [16]Huizhou Municipal Central Hospital, Sun Yat-Sen University, 516000 Huizhou, China. [17]Department of Nephrology, Beijing Tiantan Hospital, Capital Medical University, 100000 Beijing, China. [18]Department of Nephrology, Guangdong Provincial Hospital of Chinese Medicine, The Second Affiliated Hospital, The Second Clinical College, Guangzhou University of Chinese Medicine, 510000 Guangzhou, China. [19]The Third Affiliated Hospital of Southern Medical University, 510000 Guangzhou, China. [20]Institute of Health Management, Southern Medical University, 510000 Guangzhou, China. [21]DHC Technologies, 100000 Beijing, China. [22]These authors contributed equally: Changwei Wu, Yun Zhang, Sheng Nie. ✉e-mail: ffhouguangzhou@163.com; ygliu_uestc@163.com; guisenli@163.com

