## [Peer Review File · Nature Communications]

Predicting in-hospital outcomes of patients with acute kidney injuryREVIEWER COMMENTS

Reviewer #1 (Remarks to the Author):

This is a respectable machine learning attempt, using large amounts of retrospective data, from three different hospitals to attempt to predict future complications in patients with AKI. The authors reference numerous articles that have reported models to predict AKI, but few models predict hospital outcomes such as death or dialysis. One of the strengths of this paper is the fact there's a large population and it was from multiple hospitals thus enhancing the generalizability of the work. Unfortunately, they say, on numerous occasions that identifying patients at risk for these complications will improve their outcome – that is a hope rather than a proven reality; they should avoid the hyperbole.

Sadly, the authors seem to have lost track of the clinical application. It is crucially important when designing a model for clinical deployment that the clinical scenario be clear. In this case, the goal is, in a future deployment situation, at a particular moment in time to predict future complications occurring during the ensuing 24-hour epochs. At the particular moment a prediction of risk is being generated future data will not be available – as it is in a retrospective data set. Obviously, the prediction of future risk cannot rely on unavailable data at the time the prediction is made. This clinical scenario should inform the methodology, data partitioning, and selection of a machine learning algorithm that does not consider future data (temporal leakage) to make a risk assessment. The use of a bidirectional RNN at any given point in time is using future data which would not be available for clinical decision. That is the point of a bidirectional RNN and the point of this criticism. Evaluation of this model's performance would not reflect future deployment performance and would certainly be overly optimistic. This is a major flaw in the paper. If they cannot convince the readers and reviewers that this is not the case, it would be misleading by reporting an overly optimistic future deployment performance.

The paper also lacks clarity about exactly what was done and is presented in a somewhat jumbled fashion, with results appearing in the discussion and methods appearing throughout. For example, one of the examples about methods and results is seen in line 303. More methods of how the data were partitioned line 305. I do not know what the 'traversal' strategy is. These all belong in the methods.

I compliment the authors on their overall excellent English; however, it could be tightened up and shortened, such as in line 85 'novel' and 'accurately' could be deleted (excessive) as could in line 86 everything after AKI. Some hyperbole could be avoided in line 75 'is critical for' should be more tentative with replacement with 'could'. Also in line 131 'we have successfully established' should say 'we have previously reported a machine learning...' (established clinically implied is overdone).

The introduction can be shortened. Much of it belongs in (and is repeated in) the discussion. The intro basically says: 'Although there are many models that can identify patients at risk for AKI, when they develop AKI, there are few models to predict which patients will develop complications. That would be worthwhile and could perhaps guide therapy. We present an ML model to predict the complications of death and dialysis.'

A methodologic problem in this population is that the death rate is only 1.3% leaving marked class imbalance and raising the question about their diagnosis of AKI. The kidney damage must have been mild if the overall population had a lower rate of mortality than generally seen in ICU's. Perhaps it would have been better if they used a more stringent definition of AKI and were searching for targets (death dialysis) in a more severely ill population that was not quite so imbalanced. Was the incidence of dialysis reported?

The use of data from 20 past years is problematic. Is it still relevant? Medicine has certainly taken a great leap forward over those years, certainly in China. Using a temporally remote population to make predictions about future patients may not be such a great idea and certainly should at least be commented on in the paper. Also relevant to this, in performance assessment, random splitting – including data from all time points in the test set may not be optimal.

in line 207 they talk about the 'construction of sequential representation' this conversation is confusing. What I think they did was somehow combine (merge?) all the results in a 24-hour epoch and make predictions on a 24-hour basis. Was there a mathematical function used or did they use the average of all the results in the 24-hour epoch did they use the range, median, or mean? How were

features actually merged? More information is required. In addition, strangely enough, the authors are somewhat cavalier about what they did with their various models. I believe they had eight models to predict 24, 48, 72, and 7 day dichotomous outcomes of death/ no death, dialysis/ no dialysis. They actually developed and evaluated 8 models.

I am particularly curious about the statement on line 220 in this paper 'we adopted the patient data after AKI diagnosis' what does that mean?

in line 229 how were values combined, how are features extracted from these combined values, what was actually the input data for the model? There is a lot about this but I still don't quite understand precisely what was done. It would not be possible to reproduce these models and test the results or apply them clinically.

In the evaluation section the authors rather tediously tell us what F scores, precision, and recall are (unnecessarily) and leave the mention of AUROC (the most important measure of a binary discriminator which actually contains all of the information necessary to evaluate performance) to the end of the paragraph. Interestingly they do not present the precision-recall curves which some might consider are better measures with severe class imbalance. They use several methods to evaluate the performance of their model not 'effectiveness' In 280. Efficacy wasn't tested nor anything else other than model performance on the test set, as is occasionally stated throughout the paper

The partitioning methodology may be particularly important. Here it appears they randomly split their data which meant that the external test set contained patients from all 20 years. A test set should, as closely as possible, mimic the deployment scenario. The performance evaluation is only an estimate of model deployment performance – what we really want to know is how the model will perform in the clinical setting. Of course, this can't be known, it can only be estimated. Appropriate experimental design should assure that the model and test set are pertinent to that clinical setting. Hence it would be wise to use data most representative of the intended future deployment data, not data from 20 years ago. Instead of randomly splitting the data, in this setting, longitudinally by time partitioning would more accurately estimate ultimate deployment performance

In line 318 it is suddenly mentioned there are AKI stages that have not previously been described. What are these?

In line 337 through 339 there is the statement that the overall prediction performance was evaluated with F scores in AUROC and then they go on to tell us what they were in the derivation cohort. This is absolutely no estimate of future performance. AUROCs on the derivation set of 95% are expected. I really cannot imagine why they even report these results and claim that they demonstrate good performance. I want to know that the results on the test set properly designed, are a reliable estimate of future performance.

Likewise for the results in the validation cohort. The validation cohort should be used for tuning the machine learning model in an iterative process using the validation cohort to improve model performance. Therefore, one would expect model performance to improve.

Finally, after line 352, they report these the AUROCs in the actual test data set which are respectable for death, but not so precise for dialysis.

Although they mention there were comparators it is not clear until in the discussion. No mention in the methods of how this was done and the results belong, well, in the results. These just show up in the discussion where it doesn't belong. When results were compared to other models (a biLSTM and bisingle LSTM) with unacceptable performances were used but not described. I suspect that the poor performance of these or rather the good performance of the bidirectional LSTM was due to temporal leakage which would not be available in a true deployment scenario.

Ln 85 delete novel and accurately

line 224 what does 'standard time' mean?

Line 435 in their shortcomings they mentioned they did not do a subgroup analysis and now I'm curious to know why not?

line 440 the word excellent should be deleted.

Reviewer #2 (Remarks to the Author):

Wu et al developed a deep learning model based on a nationwide multicenter network that included 19 regional medical centers across China and 7,084,339 hospitalized patients, from January 2000 to May 2022 to predict the risk of in-hospital death (primary outcome) and dialysis (secondary outcome) for patients who developed AKI during hospitalization. They showed an excellent area under the curve in both internal and external validation cohorts for these outcomes.

Although the study is clinically relevant, I have several concerns with this manuscript which I have described below.

Limitations:

1. Objectives and hypothesis should be clearly stated in the introduction
2. It is not clear how the hospitals for internal and external validation cohorts were chosen. Also, how did the authors confirm that data was extracted in the same manner across the 19 hospitals. Were a small number of charts checked to validate the extracted data across all centres?
3. Exposure: I have concerns with both timing and definition of AKI used in the study. It is unclear after how many days of admission AKI occurred in this cohort. This is an important information and should have been clarified in the manuscript.
4. Also, it is unclear how AKI was defined. Authors have written KDIGO definitions were used (line 189), but a few lines later they write that definitions and stage of AKI was based on the dynamic definition described by Nanfang hospital. This should be clearly described.
5. MAKE (major adverse kidney events) is a standard way to define and understand consequences after AKI and should be used as a primary outcome. In-hospital Mortality can be a secondary outcome
6. Exclusion criteria: It is unclear whether those with kidney transplant, pre-existing CKD, previous AKI were included in the cohort.
7. Comorbid conditions of the cohort are not well described: what was the proportion of patients with obesity, sepsis, cancer, heart disease, etc. Surgery details are quite vague. Also, other illness parameters such as fluid overload and inotropic support were not mentioned.
8. Table 1: what was the proportion of missing data? How was it handled?
9. Outcomes: Mortality rate of 1.38% seems low. Previous studies have reported much higher in-patient mortality risk with AKI.
10. What modality of dialysis did the patients receive? No details were provided
11. What were the final clinical variables included in the model? Authors should clearly list these variables in the results for the readers.
12. Results of subgroup analysis were not presented. It would be good to know how models performed for various age groups, AKI stage, ICU status and presence of comorbid conditions.

Please find the point-to-point response for the comments below:

Reviewer # 1

This is a respectable machine learning attempt, using large amounts of retrospective data, from three different hospitals to attempt to predict future complications in patients with AKI. The authors reference numerous articles that have reported models to predict AKI, but few models predict hospital outcomes such as death or dialysis.

1. One of the strengths of this paper is the fact there's a large population and it was from multiple hospitals thus enhancing the generalizability of the work. Unfortunately, they say, on numerous occasions that identifying patients at risk for these complications will improve their outcome – that is a hope rather than a proven reality; they should avoid the hyperbole.

Author response: Thank you for your valuable comment. We carefully considered the wording of the entire manuscript to avoid hyperbole.

Changes made:

- Line 89-91, under Abstract, “The predictive performance was consistent in both internal and external validation cohorts. The model can predict important outcomes of patients with AKI, which could be helpful for the early management of AKI.”
 - Line 78, under Abstract, “Early prediction of AKI-related clinical events and timely intervention for high-risk patients could improve their outcomes.”
 - Line 438, under Discussion, “Despite the performance of our model when compared to those in the literature, future studies should evaluate and independently validate our model to establish its clinical utility and effects on decreasing unfavorable in-hospital and outpatient outcomes as well as to explore the role of AKIEPM in researching strategies for delivering preventive care for patients with AKI.”
 - Line 443, under Discussion, “AKIEPM could potentially become a crucial part of routine clinical pathways for AKI management.”
2. Sadly, the authors seem to have lost track of the clinical application. It is crucially important when designing a model for clinical deployment that the clinical scenario be clear. In this case, the goal is, in a future deployment situation, at a particular moment in time to predict future complications occurring during the ensuing 24-hour epochs. At the particular moment a prediction of risk is being generated future data will not be available – as it is in a retrospective

data set. Obviously, the prediction of future risk cannot rely on unavailable data at the time the prediction is made. This clinical scenario should inform the methodology, data partitioning, and selection of a machine learning algorithm that does not consider future data (temporal leakage) to make a risk assessment. The use of a bidirectional RNN at any given point in time is using future data which would not be available for clinical decision. That is the point of a bidirectional RNN and the point of this criticism. Evaluation of this model's performance would not reflect future deployment performance and would certainly be overly optimistic. This is a major flaw in the paper. If they cannot convince the readers and reviewers that this is not the case, it would be misleading by reporting an overly optimistic future deployment performance.

Author response: We appreciate your valuable comments and suggestions. As the reviewer said, it is crucially important when designing a model for clinical deployment that the clinical scenario be clear. As it is in a retrospective data set, the future data of the prediction of risk will not be available, and then the methodology, data partitioning, and selection of a machine learning algorithm should not consider future data. For the proposed method AKIEPM, we consider the clinical scenario. Although bidirectional LSTM units adopted in AKIEPM extract the features from the forward and reverse of the input sequence, their main emphasis is on analyzing the dependence among the input sequence, the forward sequence, and the reverse sequence when they are available for training. **For testing and validating the model, AKIEPM only uses the history information without future data to predict the event based on the last input sequence, which is suitable for the clinical scenario, as shown in the following figure.** For example, when training the model, for input sequence t_i , the model can use the forward sequence t_{i-1} and the reverse sequence t_{i+1} to analyze their relevance; for input sequence t_n , the model can use the forward sequence t_{n-1} but not the reverse sequence. When testing and validating the model, if we input t_1 sequence, AKIEPM predicts the event after 24h, 48h, 72h, and 7d relative to time t_1 ; if we input t_1, t_2, t_3 sequence, AKIEPM extracts the relevance among t_1, t_2, t_3 sequence and predicts the event after 24h, 48h, 72h and 7d relative to the last time t_3 , which does not use the future data. As suggested by the reviewer, we have explained the way of AKIEPM training, testing, and validating in the revised manuscript to avoid misleading of readers and reviewers. For the convenience of your review, we put

corresponding descriptions in the supplementary materials of the revised manuscript here (the italicized text).

The bidirectional LSTM units adopted in AKIEPM extract the features from the forward and reverse of the input sequence to analyze the dependence among the input sequence, the forward sequence, and the reverse sequence when they are available for training. For testing and validating the model, AKIEPM only uses the history information without future data to predict the event based on the last input sequence, which is suitable for the clinical scenario, as shown in the following Figure S1. For example, when training the model, for input sequence t_i , the model can use the forward sequence t_{i-1} and the reverse sequence t_{i+1} to analyze their relevance; for input sequence t_n , the model uses the forward sequence t_{n-1} but not the reverse sequence. When testing and validating the model, if we input t_1 sequence, AKIEPM predicts the event after 24h, 48h, 72h, and 7d relative to time t_1 ; if we input t_1, t_2, t_3 sequence, AKIEPM extracts the relevance among t_1, t_2, t_3 sequence and predicts the event after 24h, 48h, 72h and 7d relative to the last time t_3 , which does not use the future data.

- The paper also lacks clarity about exactly what was done and is presented in a somewhat jumbled fashion, with results appearing in the discussion and methods appearing throughout. For example, one of the examples about methods and results is seen in line 303. More methods of how the data were partitioned line 305.

Author response: Thank you for your comment. We have stated the objectives and hypothesis in the introduction. Besides, the methods, results, and discussion has been revised as the suggestion.

Changes made:

- Line 137-142, under Introduction, last paragraph, “Although there are many models that can identify patients at risk for AKI, however, once patients develop AKI, few models predict the risk of clinically important outcomes (such as hospital death or dialysis) in AKI patients. Predictive models for clinically important outcomes could be helpful in guiding the early management of AKI patients. Thus, we present a deep learning model to dynamically predict in-hospital death and dialysis of AKI patients.” stated the objectives and hypothesis.
- Line 213-227, Construction of derivation and validation cohorts and Figure 3 (described in Results before) were described in Methods now.
- Under Results/Prediction of outcomes in the derivation cohort previously, “We trained AKIEPM using the derivation cohort to dynamically predict the risk of in-hospital death or need for dialysis of patients with AKI at 24 h, 48 h, 72 h, and 7 d.” was been deleted.
- Under Results/Performance of prediction model in the internal validation cohort previously, “An internal validation cohort was created to validate the model, and an external validation cohort validated the model’s cross-site transportability.” was been deleted.

4. I do not know what the ‘traversal’ strategy is. These all belong in the methods.

Author response: We appreciate you for your valuable comments and suggestions. In this manuscript, we develop a deep learning model based on a nationwide multicenter cooperative network that includes 19 regional medical centers across China and 7,084,339 hospitalized patients, from January 2000 to May 2022 to dynamically predict the risk of in-hospital death and dialysis for patients who suffered from AKI during hospitalization. According to the suggestion of the reviewer, we have adjusted the structure of the manuscript. For example, we put how the data are partitioned in methods. Further, the ‘traversal’ strategy means that we enumerate different hospital combinations and sum their patient number to select the total number of one hospital combination is close to 10% ratio of total patients.

For example,

(2 hospitals) hospitals 1,2; hospitals 1,3; ...; hospitals 18,19;

(3 hospitals) hospitals 1,2,3; hospitals 1,3,4; ...; hospitals 17,18,19;

(4 hospitals) hospitals 1,2,3,4; hospitals 1,3,4,5; ...; hospitals 16,17,18,19;

...

(18 hospitals) hospitals 1,2,3,4,5,6,7,8,9,10,11,12,13,14,15,16,17,18;

hospitals 1,3,4,5,6,7,8,9,10,11,12,13,14,15,16,17,18,19;

hospitals 2,3,4,5,6,7,8,9,10,11,12,13,14,15,16,17,18,19.

5. I compliment the authors on their overall excellent English; however, it could be tightened up and shortened, such as in line 85 ‘novel’ and ‘accurately’ could be deleted (excessive) as could in line 86 everything after AKI.

Author response: Thank you for your valuable comment. Revised according to the suggestion on line 89-91, Abstract.

Changes made: The predictive performance was consistent in both internal and external validation cohorts. The model can predict important outcomes of patients with AKI, which could be helpful for the early management of AKI.

6. Some hyperbole could be avoided in line 75 ‘is critical for’ should be more tentative with replacement with ‘could’.

Author response: Thank you for your valuable comment. Revised according to the suggestion on line 78, Abstract.

Changes made: Early prediction of AKI-related clinical events and timely intervention for high-risk patients could improve their outcomes.

7. Also in line 131 ‘we have successfully established’ should say ‘we have previously reported a machine learning...’ (‘established clinically implied is overdone).

Author response: Thank you for your valuable comment. Revised according to the suggestion on line 132, Introduction, paragraph 5.

Changes made: We have previously reported a machine-learning model for AKI using a recurrent neural network (RNN) algorithm based on parameters within three days prior to hospitalization; it effectively predicted AKI and performed significantly better with time features than without it.

8. The introduction can be shortened. Much of it belongs in (and is repeated in) the discussion. The intro basically says: ‘Although there are many models that can identify patients at risk for AKI when they develop AKI, there are few models to predict which patients will develop complications. That would be worthwhile and could perhaps guide therapy. We present an ML model to predict the complications of death and dialysis.

Author response: Thank you for your comment. We have shortened the last two paragraphs into one paragraph under Introduction, last paragraph, line 137-142.

Changes made: Although there are many models that can identify patients at risk for AKI, however, once patients develop AKI, few models predict the risk of clinically important outcomes (such as hospital death or dialysis) in AKI patients. Predictive models for clinically important outcomes could be helpful in guiding the early management of AKI patients. Thus, we present a deep learning model to dynamically predict in-hospital death and dialysis of AKI patients.

9. A methodologic problem in this population is that the death rate is only 1.3% leaving a marked class imbalance and raising the question about their diagnosis of AKI. The kidney damage must have been mild if the overall population had a lower rate of mortality than generally seen in ICU's. Perhaps it would have been better if they used a more stringent definition of AKI and were searching for targets (death dialysis) in a more severely ill population that was not quite so imbalanced.

Author response: We appreciate meaningful comments and suggestions. In this study population, patients of AKI stage 2 and stage 3 only accounted for 20.13%. It might be one of the reasons for the low mortality of AKI patients in the study. All patient cohorts were derived from general hospitals, and the mortality rates were similar among different hospitals.

In order to avoid a shift in disease severity, we analyzed the subgroup of the AKI stage, whose results were shown in the following figures for predicting death and dialysis, respectively. From various evaluation indicators, it could be seen that the predictive performance of the model was well in different disease severity.

10. Was the incidence of dialysis reported?

Author response: Thanks for your questions. In Table 1 and Table 2, we both reported the incidence of dialysis in different cohorts. A total of 11205 dialysis patients accounted for 8.17% of the study population.

11. The use of data from 20 past years is problematic. Is it still relevant? Medicine has certainly taken a great leap forward over those years, certainly in China. Using a temporally remote population to make predictions about future patients may not be such a great idea and certainly should at least be commented on in the paper. Also relevant to this, in performance assessment, random splitting – including data from all time points in the test set may not be optimal.

Author response: We appreciate meaningful comments and suggestions. We extracted the data from the external validation cohort in 2019-2021 and conduct the validation experiment, whose results were shown in the following tables for predicting death and dialysis, respectively.

	Death Accuracy	AUROC	Precision	Recall	F-score
24h	0.9182	0.9158	0.7449	0.9184	0.8226
48h	0.9264	0.9263	0.7333	0.9266	0.8187
72h	0.8038	0.9078	0.8272	0.8038	0.8153
7d	0.8663	0.8721	0.7447	0.8664	0.8009

Dialysis	Accuracy	AUROC	Precision	Recall	F-score
24h	0.8573	0.7262	0.3893	0.9090	0.5451
48h	0.8277	0.7630	0.6527	0.8480	0.7377
72h	0.8652	0.7556	0.4205	0.9196	0.5771
7d	0.8481	0.6948	0.3858	0.9177	0.5432

Compared with the validation results obtained from the data from the external validation cohort in 2000-2021, the results of the data from the external validation cohort in 2019-2021 show a decrease because the data from different years have different scales. Splitting the data according to the year may result in more patient data in some years, thus, we random split to form the test set to decrease this condition. As the reviewer said, including data from all time points in the test set may not be optimal, in order to keep a quantitative balance, we adopt the random split strategy.

12. in line 207 they talk about the ‘construction of sequential representation’ this conversation is confusing. What I think they did was somehow combine (merge?) all the results in a 24-hour epoch and make predictions on a 24-hour basis. Was there a mathematical function used or did they use the average of all the results in the 24-hour epoch did they use the range, median, or mean? How were features actually merged? More information is required.

Author response: We appreciate you for your constructive comments and suggestions. Because the clinical record of patients is conducted on the actual occurrence time. For example, the detection time of blood pressure of a patient is 2020.1.1 9:10:00 (Record A), but the detection time of creatinine of a patient is 2020.1.1 10:12:45 (Record B). If we construct the sequential representation at actual time, Record A only records blood pressure without other variables, the same as Record B. Thus, we combine the records in 24 hours to decrease missing values according to merge rules, like Record AB can record the values of blood pressure and creatinine by merging. We do not use mathematical functions and the range, median, or mean, we directly use a value that already exists according to merge rules. To make the reviewer understand how to merge, we give some examples with merge rules as follows.

The merge rules were as follows, wherein we took the detection time of creatinine as the standard time (Figure 4):

Figure 4 Schematic of the merge rules in the AKI event prediction model AKIEPM.

- (1) If there was no SCr result within 24 h, all laboratory values were combined with that record at the latest SCr measurement. If there were multiple values for the same indicator, the latest value was taken.

Time	temperature	pulse	respiration	sbp	dbp	SCr	pct	wbc
2020.1.1 9:10:00	36.1	70	20	108	61			
2020.1.1 10:12:45							4.46	40.30
2020.1.1 11:13:20								25.27
2020.1.1 11:13:20(Merged)	36.1	70	20	108	61	-	4.46	25.27

- (2) If there was one creatinine value within 24 h, other laboratory values were combined with that record at the time of SCr measurement. If there were multiple values for the same indicator, those simultaneous with the SCr measurement or the closest were taken.

- a. If there were multiple values, the maximum temperature, lowest (<50) and highest (>50) pulse rates, minimum blood pressure, maximum respiration rate, and laboratory values closest to the time of SCr measurement were taken.

Time	temperature	pulse	respiration	sbp	dbp	SCr	pct	wbc
2020.1.1 9:10:00	36.1	70	20	108	61			
2020.1.1 10:12:45	36.0	69	22			69	4.46	40.30
2020.1.1 11:13:20								25.27
2020.1.1 10:12:45(Merged)	36.1	70	22	108	61	69	4.46	40.30

- b. If there were multiple values for the operation, medications, prescriptions, diagnosis, and events per day, we obtained those closest to when the SCr levels was measured.

Time	operation	medications	prescriptions	diagnosis	events	SCr
2020.1.1 9:10:00	op_A				eve_A	
2020.1.1 10:12:45	op_B	med_A				69
2020.1.1 11:13:20		med_B			eve_B	

2020.1.1 10:12:45(Merged)	op _A ; op _B	med _A ; med _B	-	-	eve _A ; eve _B	69
-----------------------------------	-------------------------------------	---	---	-------------------------------------	----

(3) If there were two SCr values within 24 h, laboratory values before and after the first SCr measurement were combined with the first and second SCr measurements, respectively. If there were multiple values for other laboratory indicators, refer to (2).

13. In addition, strangely enough, the authors are somewhat cavalier about what they did with their various models. I believe they had eight models to predict 24, 48, 72, and 7-day dichotomous outcomes of death/ no death, dialysis/ no dialysis. They actually developed and evaluated 8 models.

Author response: We appreciate you for your constructive comments and suggestions. As the reviewer said, we first develop 8 models to predict 24, 48, 72, and 7d outcomes of death/no death, dialysis/no dialysis, and then we integrate them into AKIEPM. According to the comments of the reviewer, we have pointed out the above idea in the section of the prediction model of the revised manuscript.

14. I am particularly curious about the statement on line 220 in this paper ‘we adopted the patient data after AKI diagnosis’ what does that mean?

Author response: We appreciate you for your constructive comments and suggestions. “we adopted the patient data after AKI diagnosis” means that the model uses the patient data after AKI diagnosis. For example, if a patient is hospitalized since January 1, 2020, but he is diagnosed with AKI on January 10, 2020, then we adopt his data after January 10, 2020.

15. in line 229 how were values combined, how are features extracted from these combined values, and what was actually the input data for the model? There is a lot about this but I still don't quite understand precisely what was done. It would not be possible to reproduce these models and test the results or apply them clinically.

Author response: We appreciate you for your constructive comments and suggestions. We combine the data according to merge rules by the value already existing. To make the reviewer understand how to merge, we give some examples with merge rules as follows. The merge

rules were as follows, wherein we took the detection time of creatinine as the standard time (Figure 4):

Figure 4 Schematic of the merge rules in the AKI event prediction model AKIEPM.

- (1) If there was no SCr result within 24 h, all laboratory values were combined with that record at the latest SCr measurement. If there were multiple values for the same indicator, the latest value was taken.

Time	temperature	pulse	respiration	sbp	dbp	SCr	pct	wbc
2020.1.1 9:10:00	36.1	70	20	108	61			
2020.1.1 10:12:45							4.46	40.30
2020.1.1 11:13:20								25.27
2020.1.1 11:13:20(Merged)	36.1	70	20	108	61	-	4.46	25.27

- (2) If there was one creatinine value within 24 h, other laboratory values were combined with that record at the time of SCr measurement. If there were multiple values for the same indicator, those simultaneous with the SCr measurement or the closest were taken.
- a. If there were multiple values, the maximum temperature, lowest (<50) and highest (>50) pulse rates, minimum blood pressure, maximum respiration rate, and laboratory values closest to the time of SCr measurement were taken.

Time	temperature	pulse	respiration	sbp	dbp	SCr	pct	wbc
2020.1.1 9:10:00	36.1	70	20	108	61			
2020.1.1 10:12:45	36.0	69	22			69	4.46	40.30
2020.1.1 11:13:20								25.27
2020.1.1 10:12:45(Merged)	36.1	70	22	108	61	69	4.46	40.30

b. If there were multiple values for the operation, medications, prescriptions, diagnosis, and events per day, we obtained those closest to when the SCr levels were measured.

Time	operation	medications	prescriptions	diagnosis	events	SCr
2020.1.1 9:10:00	op_A				eve_A	
2020.1.1 10:12:45	op_B	med_A				69
2020.1.1 11:13:20		med_B			eve_B	
2020.1.1 10:12:45(Merged)	op _A ; op _B	med _A ; med _B	-	-	eve _A ; eve _B	69

(3) If there were two SCr values within 24 h, laboratory values before and after the first SCr measurement were combined with the first and second SCr measurements, respectively. If there were multiple values for other laboratory indicators, refer to (2).

The sequential representation t^p is input into the model, that is, the merged data of patients, for example, the merged data in above three tables are input into the model. Then the bidirectional LSTM is used to extract the hidden features for the embedding of sequential representation based on input, forget, and output units i_t, f_t, o_t of LSTM at time t with

$$i_t = f(W_{ie}E_t^i + W_{ih}h_{t-1} + w_{ic} \circ c_{t-1} + b_i),$$

$$f_t = f(W_{fe}E_t^i + W_{fh}h_{t-1} + w_{fc} \circ c_{t-1} + b_f),$$

$$c_t = f_t \circ c_{t-1} + i_t \circ g(W_{ce}E_t^i + W_{ch}h_{t-1} + b_c),$$

$$o_t = f(W_{oe}E_t^i + W_{oh}h_{t-1} + w_{oc} \circ c_{t-1} + b_o),$$

$$h_t = o_t \circ g(c_t).$$

Here, c_t is cell state, E_t^i and h_t are the input and hidden representations, respectively. $f()$ adopts sigmoid function and $g()$ adopts tanh nonlinear function with weight $W_{ie}, W_{fe}, W_{ce}, W_{oe}, W_{ih}, W_{fh}, W_{ch}$ and W_{oh} connecting different inputs and gates with memory cells and outputs, as well as biases b_i, b_f, b_c and b_o . Cell state ct is updated with a fraction of the previous cell state c_{t-1} controlled by f_t , and a new input state is created from the element-wise product \circ of i_t and the output of the cell state by $g()$ to remember the related features of death

risk of AKI inpatients from historical records and current state. The peephole connection weights w_{ic} , w_{fc} and w_{oc} further influence the input, forget, and output units.

16. In the evaluation section the authors rather tediously tell us what F scores, precision, and recall are (unnecessarily) and leave the mention of AUROC (the most important measure of a binary discriminator which actually contains all of the information necessary to evaluate performance) to the end of the paragraph.

Author response: Thank you for your comment. Suggestions have been incorporated under Methods/ Evaluation, line 288.

Changes made: We adopted precision, recall, F-score, and area under the receiver operator curve (AUROC) to evaluate the performance of the proposed model.

17. Interestingly they do not present the precision-recall curves which some might consider are better measures with severe class imbalance.

Author response: Thank you for your comment. For the class imbalance problem, we adopt a random oversampling strategy to decrease the imbalance problem, then we do not show the precision-recall curves. According to the suggestions of the reviewer, we have added corresponding descriptions in the section on the partition of derivation, internal validation, and external validation cohorts.

18. They use several methods to evaluate the performance of their model not ‘effectiveness’ In 280. Efficacy wasn't tested nor anything else other than model performance on the test set, as is occasionally stated throughout the paper.

Author response: We appreciate you for your meaningful comments and suggestions. Reference Tomašev et al. [1], we adopted the Precision, Recall and AUROC to evaluate the performance of AKIEPM. Meanwhile, we added F-score, which was the comprehensive measurement of Precision and Recall. In addition, we used Accuracy to assist evaluate the performance. We evaluated the model in the derivation, internal validation, and external validation cohorts by Precision, Recall, F-score, AUROC, and Accuracy, respectively.

[1] Tomašev et al. Use of deep learning to develop continuous-risk models for adverse event prediction from electronic health records [J]. Nature Protocols, 16: 2765-2787, 2021.

19. The partitioning methodology may be particularly important. Here it appears they randomly split their data which meant that the external test set contained patients from all 20 years. A test set should, as closely as possible, mimic the deployment scenario. The performance evaluation is only an estimate of model deployment performance – what we really want to know is how the model will perform in the clinical setting. Of course, this can't be known, it can only be estimated. Appropriate experimental design should assure that the model and test set are pertinent to that clinical setting. Hence it would be wise to use data most representative of the intended future deployment data, not data from 20 years ago. Instead of randomly splitting the data, in this setting, longitudinally by time partitioning would more accurately estimate ultimate deployment performance.

Author response: We appreciate you for your meaningful comments and suggestions. We extract the data from the external validation cohort in 2019-2021 and conduct the validation experiment, whose results are shown in the following tables for predicting death and dialysis, respectively.

Death	Accuracy	AUROC	Precision	Recall	F-score
24h	0.9182	0.9158	0.7449	0.9184	0.8226
48h	0.9264	0.9263	0.7333	0.9266	0.8187
72h	0.8038	0.9078	0.8272	0.8038	0.8153
7d	0.8663	0.8721	0.7447	0.8664	0.8009

Dialysis	Accuracy	AUROC	Precision	Recall	F-score
24h	0.8573	0.7262	0.3893	0.9090	0.5451
48h	0.8277	0.7630	0.6527	0.8480	0.7377
72h	0.8652	0.7556	0.4205	0.9196	0.5771
7d	0.8481	0.6948	0.3858	0.9177	0.5432

Compared with the validation results obtained from the data from the external validation cohort in 2000-2021, the results of the data from the external validation cohort in 2019-2021 show a decrease because the data from different years have different scales. Splitting the data according to the year may result in more patient data in some years, thus, we random split to form the test set to decrease this condition. As the reviewer said, including data from all time

points in the test set may not be optimal, in order to keep a quantitative balance, we adopt the random split strategy.

20. In line 318 it is suddenly mentioned there are AKI stages that have not previously been described. What are these?

Author response: Thanks for your questions. We have described the definition of the AKI stage under Methods/Definitions, line 198-200.

Changes made: AKI stages were determined by the peak SCr level after AKI detection, with a rise of less than 100% indicating stage 1, a rise of 100% or more indicating stage 2, and a rise of 200% or more over baseline indicating stage 3.

21. In line 337 through 339 there is the statement that the overall prediction performance was evaluated with F scores in AUROC and then they go on to tell us what they were in the derivation cohort. This is absolutely no estimate of future performance. AUROCs on the derivation set of 95% are expected. I really cannot imagine why they even report these results and claim that they demonstrate good performance. I want to know that the results on the test set properly designed, are a reliable estimate of future performance.

Author response: We appreciate you for your constructive comments and suggestions. In the evaluation, we use precision, recall, and f-score to evaluate the performance of the proposed model. We also use AUROC to compare the overall prediction performance. In addition, accuracy is used to assist in judging performance. As the reviewer said, AUROC on the derivation set of 95% is expected. We also report the results in internal validation (test set) and external validation cohorts (validation set). The internal validation cohort's AUROCs for predicting death and need for dialysis were 93.58%, 92.45%, 93.02%, and 87.03%, and 88.33%, 82.73%, 83.09%, and 77.25% at 24 h, 48 h, 72 h, and 7 d, respectively. In the external validation cohort, the AUROCs for death and need for dialysis were 92.43%, 92.16%, 88.36%, and 88.32%, and 74.18%, 77.58%, 75.21%, and 69.34% at 24 h, 48 h, 72 h, and 7 d, respectively. Because the external validation cohort comes from three independent hospitals, we consider the results in external validation cohort can be seen as an estimate of future prediction performance. In the future, we will deploy our model in actual hospitals to obtain actual prediction performance.

22. Likewise for the results in the validation cohort. The validation cohort should be used for tuning the machine learning model in an iterative process using the validation cohort to improve model performance. Therefore, one would expect model performance to improve.

Author response: We appreciate you for your constructive comments and suggestions. In the proposed, we use internal validation to tune the model in an iterative process to improve model performance. Because the external validation cohort comes from three independent hospitals, we consider the results in the external validation cohort can be seen as an estimate of future prediction performance, then we use the external validation cohort to verify the prediction performance of AKIEPM. On the other hand, AKIEPM is training and testing on the derivation and internal validation cohort, then it gets poorer but acceptable results on the external validation cohort.

23. Finally, after line 352, they report these the AUROCs in the actual test data set which are respectable for death, but not so precise for dialysis.

Author response: Thank you for your comment. The performance in predicting the need for dialysis was indeed lower than that of death. One possible reason was that different physicians make different decisions about whether and when to initiate dialysis for a patient based on factors, including disease status, personal willingness, and accessibility to dialysis.

24. Although they mention there were comparators it is not clear until the discussion. No mention in the methods of how this was done and the results belong, well, in the results. These just show up in the discussion where it doesn't belong. When results were compared to other models (a biLSTM and bisingle LSTM) unacceptable performances were used but not described. I suspect that the poor performance of these or rather the good performance of the bidirectional LSTM was due to temporal leakage which would not be available in a true deployment scenario.

Author response: We appreciate you for your constructive comments and suggestions. We compare the proposed method with two methods BiLSTM and BiSingleLSTM, which are conducted by training on the same derivation cohort, testing on the same internal validation cohort, and validating on the external validation cohort. According to the suggestions of the

reviewer, we have added the comparing methods BiLSTM and BiSingleLSTM in the Evaluation section. For clearing understanding them, we give a table to explain the technology used by them as follows.

	category embedding	bidirectional LSTM	one-way LSTM	softmax
AKIEPM	yes	yes	yes	yes
BiLSTM	no	yes	no	yes
BiSingleLSTM	no	yes	yes	yes

Although AKIEPM extracts the features from the forward and reverse of the input sequence, its main emphasis is on analyzing the dependence among the input sequence, the forward sequence, and the reverse sequence when they are available for training. For testing and validating the model, AKIEPM only uses the history information without future data to predict the event based on the last input sequence, which is suitable for the clinical scenario, as described in the response to the first comment. We consider that BiLSTM and BiSingleLSTM gain poor performance may be because they do not encode the category to capture the meanings of clinical variables or they are stuck in local optimality and stop early. According to the comments of the reviewer, we have added the above description in the revised manuscript.

Changes made:

- Line 400-406, under Discussion, the third paragraph, “To further verify the performance of the model, we used BiLSTM and BiSingleLSTM 31 as a comparison. In our model, category embedding and one-way LSTM were added to fuse the hidden representation at all times. Based on our results, the performance of BiLSTM and BiSingleLSTM was worse than our model. We considered that BiLSTM and BiSingleLSTM gained poor performance might be because they do not encode the category to capture the meanings of clinical variables or they were stuck in local optimality and stopped early.”
- Line 356-358, under Results/ Comparing with baseline algorithms, “The results of baseline comparison were provided in Supplementary Table 2. Whether predicting death or dialysis, BiLSTM and BiSingleLSTM both had relatively poor predictive capacity.”

25. Ln 85 delete novel and accurately

Author response: Thanks for your suggestion. Revised according to the suggestion on line 90, Abstract.

Changes made: The model can predict important outcomes of patients with AKI, which could be helpful for the early management of AKI.

26. line 224 what does 'standard time' mean?

Author response: Thank you for your question. The standard time defined in the manuscript means the detection time of creatinine in patients.

27. Line 435 in their shortcomings they mentioned they did not do a subgroup analysis and now I'm curious to know why not?

Author response: Thank you for your comment. We have shown the results of subgroup analysis in Results/ Subgroup analysis of prediction model, line 360-370.

Changes made: We also conducted a subgroup analysis of previously reported risk factors related to the mortality or dialysis, including, age, gender, hypertension, diabetes, AKI stage, baseline SCr, length of ICU stay, and major surgery. Figure 7 had shown the AUROC curve at 24-hour, 28-hour, 72-hour, and 7-day for predicting death and dialysis. All AUROCs of various subgroups were almost more than 80% in predicting death and more than 75% in predicting dialysis, which indicated that AKIEPM performed well in the above-mentioned clinical situations.

Figure 7 Prediction of death and dialysis in Subgroup cohorts. X-axis was the value of AUROC, and Y-axis was the model predicting death and dialysis at 24-hour, 28-hour, 72-hour, and 7-day.

28. line 440 the word excellent should be deleted.

Author response: Thanks for your suggestion. Revised according to the suggestion on line 438, Discussion.

Changes made: Despite the performance of our model when compared to those in the literature, future studies should evaluate and independently validate our model to establish its clinical utility and effects on decreasing unfavorable in-hospital and outpatient outcomes as well as to explore the role of AKIEPM in researching strategies for delivering preventive care for patients with AKI.

Reviewer # 2

Wu et al developed a deep learning model based on a nationwide multicenter network that included 19 regional medical centers across China and 7,084,339 hospitalized patients, from January 2000 to May 2022 to predict the risk of in-hospital death (primary outcome) and dialysis (secondary outcome) for patients who developed AKI during hospitalization. They showed an excellent area under the curve in both internal and external validation cohorts for these outcomes.

Although the study is clinically relevant, I have several concerns with this manuscript which I have described below.

Limitations:

1. Objectives and hypothesis should be clearly stated in the introduction.

Author response: Thank you for your comment. We have stated the objectives and hypothesis under Introduction, last paragraph, line 137-142.

Changes made: Although there are many models that can identify patients at risk for AKI, however, once patients develop AKI, few models predict the risk of clinically important outcomes (such as hospital death or dialysis) in AKI patients. Predictive models for clinically important outcomes could be helpful in guiding the early management of AKI patients. Thus, we present a deep learning model to dynamically predict in-hospital death and dialysis of AKI patients.

2. It is not clear how the hospitals for internal and external validation cohorts were chosen.

Author response: We appreciate you for your constructive comments and suggestions. To train and validate the performance of the prediction model, we partitioned the patients into the derivation, internal validation, and external validation cohorts with a 7:2:1 ratio. We first construct the external validation cohort, we adopt a traversal strategy to enumerate different combinations of hospitals and chose one hospital combination wherein the number of patients was closest to the 10% of the overall cohort (14,610 patients). That is, we enumerate different hospital combinations and sum their patient number to select the total number of one hospital combination that is close to 10% ratio of total patients.

For example,

(2 hospitals) hospitals 1,2; hospitals 1,3; ...; hospitals 18,19;

(3 hospitals) hospitals 1,2,3; hospitals 1,3,4; ...; hospitals 17,18,19;

(4 hospitals) hospitals 1,2,3,4; hospitals 1,3,4,5; ...; hospitals 16,17,18,19;

...

(18 hospitals) hospitals 1,2,3,4,5,6,7,8,9,10,11,12,13,14,15,16,17,18;

hospitals 1,3,4,5,6,7,8,9,10,11,12,13,14,15,16,17,18,19;

hospitals 2,3,4,5,6,7,8,9,10,11,12,13,14,15,16,17,18,19.

If the total patient number of a hospital combination is close to 10% of the overall cohort, we select them. Then we randomly select approximately 22% of the patients (27,217 patients) to form the internal validation cohort. Finally, the rest of the overall cohort forms a derivation cohort.

3. Also, how did the authors confirm that data was extracted in the same manner across the 19 hospitals. Were a small number of charts checked to validate the extracted data across all centres?

Author response: Thanks for your questions. Our data collection team used uniform standard tables and processed data at each center and conducted rigorous quality control after data collection. The standard tables were shown in Supplemental Materials, Table S3-S12.

4. Exposure: I have concerns with both timing and definition of AKI used in the study. It is unclear after how many days of admission AKI occurred in this cohort. This is an important information and should have been clarified in the manuscript.

Author response: Thanks for your questions. We have described the definition of the date of AKI under Methods/Definitions, line 195.

Changes made: The date of AKI diagnosis was defined as the earliest day on which the SCr change met the KDIGO criteria.

5. Also, it is unclear how AKI was defined. Authors have written KDIGO definitions were used (line 189), but a few lines later they write that definitions and stage of AKI was based on the dynamic definition described by Nanfang Hospital. This should be clearly described.

Author response: Thanks for your questions. We have described the definition of AKI and the AKI stage under Methods/Definitions, line 187-200.

Changes made: AKI was defined according to the Kidney Disease Improving Global Outcomes (KDIGO) clinical practice guideline. Due to the lack and accuracy of urine output, we employed SCr to define AKI as an increase by 0.3 mg/dL within 48 hours or 50% increase from baseline within 7 days. The definition of AKI was based on the dynamic criteria described by Nanfang Hospital. The SCr data collected during hospitalization were sorted in increasing order by sequential test time. For any time point t, a baseline average SCr was dynamically defined as the average value of SCr within 7 days before the point t, then each available SCr value within 7 days after point t was compared with this baseline average SCr. The date of AKI diagnosis was defined as the earliest day on which the SCr change met the KDIGO criteria. The baseline value for further analysis in patients with AKI was defined as the value at the time of AKI diagnosis. AKI stages were determined by the peak SCr level after AKI detection, with a rise of less than 100% indicating stage 1, a rise of 100% or more indicating stage 2, and a rise of 200% or more over baseline indicating stage 3.

6. MAKE (major adverse kidney events) is a standard way to define and understand consequences after AKI and should be used as a primary outcome. In-hospital Mortality can be a secondary outcome.

Author response: Thank you for your suggestions. Previous literature showed that MAKE was the composite of the first occurrence of a sustained 40% decrease in eGFR, the development of kidney failure (defined as a need for maintenance dialysis or kidney transplant and adjudicated), or death due to kidney disease (defined as death due to kidney failure that required dialysis and that could be avoided by timely dialysis). However, the confirmation of the important clinical outcomes of MAKE requires a definitive period of follow-up, such as 28 or 90 days. Unfortunately, our study is based on a dataset that lacks follow-up data after discharge. Therefore, we chose important clinical events in the hospital (death and dialysis) as the key outcomes of in our study.

7. Exclusion criteria: It is unclear whether those with kidney transplant, pre-existing CKD, previous AKI were included in the cohort.

Author response: Thanks for your questions. We have improved the exclusion criteria under the Methods/ Data description, line 178.

Changes made: The exclusion criteria were as follows: (a) patients who had less than two serum creatinine (SCr) results during hospitalization; (b) patients <18 years old; (c) patients who had HIV; and (d) patients who had end-stage kidney disease (ESKD, defined as maintenance dialysis, kidney transplantation, or eGFR <15 ml/min per 1.73 m²).

8. Comorbid conditions of the cohort are not well described: what was the proportion of patients with obesity, sepsis, cancer, heart disease, etc. Surgery details are quite vague. Also, other illness parameters such as fluid overload and inotropic support were not mentioned.

Author response: Thanks for your questions. In this study, we used the Charlson comorbidity score to calculate the burden of comorbidity. In Table 1 and Table 2, we described the Charlson comorbidity score and the proportion of patients with hypertension and diabetes. Due to the database limitation, we do not have the data about fluid overload and inotropic support. For surgery, we used the surgical grading system, which was used in China to classify the procedure from levels 1 to 4 based on difficulty, complexity, and risk. In this study, major surgery was defined as grade 4 surgery, as well as grade 3 invasively surgical operation in major body cavities (heart, intracranial, chest, abdomen, or pelvic).

9. Table 1: what was the proportion of missing data? How was it handled?

Author response: Thanks for your questions. The proportion of missing data was listed in the table as follows. In this study, we analyzed by removing missing data.

factors	Scr	BUN	BUN/Scr	Cystatin C	eGFR	Proteinuria	GLU	ALB
Missing (%)	0.51	1.47	58.12	50.91	0.51	37.11	7.18	4.44
factors	ALT	AST	TBIL	DBIL	WBC	HB	PLT	NLR
Missing (%)	4.53	4.96	6.75	9.30	2.88	7.32	2.89	3.65
factors	CRP	ESR	PCT	APTT	D-dimer	BNP	K	Na
Missing (%)	68.56	85.28	57.23	14.46	36.44	87.08	0.85	0.85
factors	Ca	Cl	P					
Missing (%)	78.70	1.05	28.52					

10. Outcomes: Mortality rate of 1.38% seems low. Previous studies have reported much higher in-patient mortality risk with AKI.

Author response: We appreciate your meaningful comments and suggestions. In this study population, patients of AKI stage 2 and stage 3 only accounted for 20.13%, which was the possible reason for the lower rate of mortality. In order to avoid a shift in disease severity, we analyzed the subgroup of the AKI stage, whose results were shown in the following figures for predicting death and dialysis, respectively. From various evaluation indicators, it can be seen that the predictive performance of the model was well in different disease severity.

11. What modality of dialysis did the patients receive? No details were provided.

Author response: Thanks for your questions. We have described the definition of dialysis under Methods/Definitions, line 203.

Changes made: In this study, dialysis included temporary or maintenance hemodialysis and peritoneal dialysis, and continuous renal replacement therapy (CRRT).

12. What were the final clinical variables included in the model? Authors should clearly list these variables in the results for the readers.

Author response: Thank you for your comment. The final clinical variables included in the model were listed in Table S1 in the Supplemental Manuscript.

Table S1 Index included in predictive mode

	index
Demographic characteristics	Age, gender
Clinical characteristics	MAP, SBP, DBP, AKI stage, comorbidities
Operation	ICU, mechanical ventilation, surgery, length of ICU, and mechanical ventilation
Treatment	drug
Experiment characteristics	Alb, ALT, APTT, AST, BUN, BNP, BUN/SCr, Ca, Cl, CRP, cTn, Cys-C, DBIL, D-Dimer, eGFR, ESR, FDP, FIB, Glu, HB, INR, LDH, NLR, Na, P, PCT, PLT, proteinuria, 24 proteinuria, PT, RBC, SCr, TBIL, TC, TCa, TG, TP, TT, UA, WBC, Blood gas analysis factors.

13. Results of subgroup analysis were not presented. It would be good to know how models performed for various age groups, AKI stage, ICU status and presence of comorbid conditions.

Author response: Thank you for your comment. We have shown the results of subgroup analysis in Results/ Subgroup analysis of prediction model, line 360-370.

Changes made: We also conducted a subgroup analysis of previously reported risk factors related to the mortality or dialysis, including, age, gender, hypertension, diabetes, AKI stage, baseline SCr, length of ICU stay, and major surgery. Figure 7 had shown the AUROC curve at 24-hour, 28-hour, 72-hour, and 7-day for predicting death and dialysis. All AUROCs of various subgroups were almost more than 80% in predicting death and more than 75% in predicting dialysis, which indicated that AKIEPM performed well in the above-mentioned clinical situations.

Figure 7 Prediction of death and dialysis in Subgroup cohorts. X-axis was the value of AUROC, and Y-axis was model predicting death and dialysis at 24-hour, 48-hour, 72-hour, and 7-day.

REVIEWERS' COMMENTS

Reviewer #1 (Remarks to the Author):

Thank you for clarifying a great many points. I have still have reservations about the clinical applicability of his complex machine learning approach. I don't think it can be applied from this paper to uses outside your hospital groups and obviously this complex data management will not be readily reputable by others.

I have a little difficulty understanding your comments about scale, by which you mean the number of patients in each year I really don't understand why that prevented longitudinal data splitting temporally. Could you explain that a little further, please .

I do think that we do not have a good idea of how this model will perform in a current population. This needs to be absolutely delineated. As a shortcoming of this paper.

In addition, you continue to imply that your model will improve outcomes there is no evidence of this. The clinical trial of this algorithm has not been done.

Reviewer #2 (Remarks to the Author):

Authors have answered all my questions satisfactorily. I have no further comments.

Please find the point-to-point response for the comments below:

Reviewer #1

1. Thank you for clarifying a great many points. I have still have reservations about the clinical applicability of his complex machine learning approach. I don't think it can be applied from this paper to uses outside your hospital groups and obviously this complex data management will not be readily reputable by others.

Author response: Thank you for your comment. The data used in this model are taken from the HIS system widely used in clinical practice and include demographics, diagnoses, prescriptions, laboratory test results, etc. These are routine data generated during clinical treatment and do not necessitate any special collection. As a result, the data set chosen for this study is routinely obtained in each hospital but only covers a broad time span.

Original data were first deidentified, inspected and then exported from the electronic hospitalization information system of each center. The data were pooled, standardized and cleaned, followed by structural processing and quality control procedures. Data cleaning is a routine preprocessing method for data analysis, and our cleaning method has been standardized into tables presented in Table S3-S12 so that it can be used for other different hospitals.

After developing the model and completing internal validation, we validated it in an external validation group comprised of three independent hospitals to assess the model's cross-site transportability. The results demonstrated that this model's cross-site transportability performed well.

Indeed, clinical interventions involving AI should undergo a rigorous and prospective evaluation to demonstrate their impact on health outcomes, which was our limitation and we also specifically pointed out this limitation in the Discussion Section. Our plan is to conduct a prospective controlled study to assess and independently validate whether our model can improve adverse outcomes in AKI

patients. This work is currently being prepared, and we will release the research findings once the trial is completed.

Changes made:

- Line 288-289, under last paragraph of Discussion, “Those who were at risk for AKI could be identified through AKIEPM.”
- Line 277-279, under Discussion, “AKIEPM was conducted on previous data, and whether it can help improve prognosis must be confirmed in prospective controlled study.”

2. I have a little difficulty understanding your comments about scale, by which you mean the number of patients in each year I really don't understand why that prevented longitudinal data splitting temporally. Could you explain that a little further, please.

Author response: We appreciate you for your valuable comments and suggestions. We are sorry for making confusion about the scale. According to the comments and suggestions of the reviewer, we explain this from following perspectives:

- ① The influence of splitting longitudinal data temporally for scale.
 - ② The estimation of ultimate deployment performance.
- ① **The influence of splitting longitudinal data temporally for scale.** As the reviewer said, the scale means the number of patients in each year. For example, there are the distribution of patient numbers in different years, as shown in the following figure.

We can find that the patients who meet inclusion and exclusion criteria are different in different years. Then, splitting the data according to time may result in more patient data in some years, this is, the data in different years have different scales. In our study, there are one derivation cohort and one internal validation cohort by random split strategy. Splitting the data according to time will produce multiply derivation cohorts and internal validation cohorts (splitting by year).

For training the model on multiply derivation data, the model may overfit to learn the hidden features of certain years, that is, it may perform well in corresponding years and predict poor in other years. To keep a quantitative balance of the data in different years, we adopt random split strategy to form one derivation cohort, in other words, we include the patients at different times of admission, and train the model in one derivation cohort.

For testing on multiply internal validation data (the same as external validating), their performance may be difficult to evaluate consistently (Situation 3 in the following example). For example, if we split the data by time, we can obtain multiply test data, such as A and B. There are three situations as follows.

(1) if the scale and result in A are both greater than those in B (e.g., $10000 > 2000$ and $80\% > 70\%$), then we can consider the model in A performs well.

A	2020	10000	80%	2021	2000	70%	B
---	------	-------	-----	------	------	-----	---

(2) if the scale and result in A are both less than those in B (e.g., $2000 < 10000$ and $70\% < 80\%$), then we can consider the model in B performs well.

A	2020	2000	70%	2021	10000	80%	B
---	------	------	-----	------	-------	-----	---

(3) if the scale in A is greater than it in B, but the result is less than it in B (e.g., $10000 > 2000$ and $70\% < 80\%$), then we differently evaluate that the model performs well in A or B for two factors (scale and result).

A	2020	10000	70%	2021	2000	80%	B
---	------	-------	-----	------	------	-----	---

Splitting the data according to the year will produce multiply test data, then it exists above situation. In one iteration, we differently evaluate the model performance among multiply test data. In multiple iterations, we can evaluate the model performance with identical one test data (scale is the same). In our manuscript, we adopt random split strategy to form one internal validation cohort. Since there is only one internal validation cohort, we can compare the results of different iterations (scale is the same) to choose the model with the best performance.

We very appreciate the reviewer for your suggestions, which give us inspiration. We consider that there are two strategies to train, test and validate the model on the data splitting by time (Please refer to Appendix A in this document). **By comprehensive consideration, these strategies in Appendix A could increase the complexity of model. Meanwhile, one derivation cohort and one internal validation cohort splitting by random split strategy can have larger data volume, which enables deep learning model with better generalization, then we choose random split strategy in our manuscript.**

② **The estimation of ultimate deployment performance.** The scale does not prevent longitudinal data splitting temporally. As the reviewer said, a test set should mimic the deployment scenario. The longitudinally by time partitioning would more accurately estimate ultimate deployment performance. It would be wise to use data most representative of the intended future deployment data. We

very appreciate the reviewer for your suggestions, which give us inspiration for estimating the performance in deployment.

In last revision, the time of external verification cohort (one set) concentrates on 2016-2020 (the time distribution is shown in the following figure), which is close to the current time, with respect to derivation and internal validation cohorts. According to the suggestions of the reviewer, we can consider that the performance in deployment can be estimated by the performance in external verification cohort to a certain extent. (We have explained this in the Evaluation section of the revised manuscript. For the convenience of your review, we put corresponding descriptions in the supplementary materials of the revised manuscript here (the italicized text). The time of external verification cohort concentrates on 2016-2020, which is close to the current time, with respect to derivation and internal validation cohorts, we can consider that the performance in ultimate deployment can be estimated by the performance in external verification cohort to a certain extent.)

Meanwhile, we extract the data from the external validation cohort in 2019-2021 (one set), which is close to the current time, and conduct the validation experiment, whose results are shown in the following tables for predicting death and dialysis, respectively.

Death	Accuracy	AUROC	Precision	Recall	F-score
24h	0.9182	0.9158	0.7449	0.9184	0.8226
48h	0.9264	0.9263	0.7333	0.9266	0.8187
72h	0.8038	0.9078	0.8272	0.8038	0.8153
7d	0.8663	0.8721	0.7447	0.8664	0.8009

Dialysis	Accuracy	AUROC	Precision	Recall	F-score
24h	0.8573	0.7262	0.3893	0.9090	0.5451
48h	0.8277	0.7630	0.6527	0.8480	0.7377
72h	0.8652	0.7556	0.4205	0.9196	0.5771
7d	0.8481	0.6948	0.3858	0.9177	0.5432

The results of the data from the external validation cohort in 2019-2021 are close to the results in the total external validation cohort, for example, the AUROC for predicting death within 24h and 48h is 91.58% and 92.63%, respectively, which shows the proposed method also obtains better performance. Then we can estimate the performance in deployment to a certain extent by these evaluations.

Changes made: Page 18, Line 415-426 and Page 19, Line 433-440

- I do think that we do not have a good idea of how this model will perform in a current population. This needs to be absolutely delineated. As a shortcoming of this paper.

Author response: Thank you for your constructive comments and suggestions. We are sorry for making confusion about how this model will perform in current population. AKIEPM is an end-to-end model, according to the comments and suggestions of the reviewer, we first introduce each layer of the proposed method AKIEPM in details (Please refer to Appendix B in this document and Supplemental Manuscript), then we conduct this model step by step with an example for better understanding the process (Please refer to Appendix C in this document). Based on above introduction, we analyze the processing of AKIEPM carefully and consider that **there maybe four reasons to make the proposed method perform well in current population.**

1) The embedding layer maps the clinical variables to continuous vector, which enables the feature extraction layer to better capture its related features.

2) The bidirectional LSTM units in feature extraction layer extract the related features of outcomes from the forward and reverse of the input sequence to analyze their dependences, which consider the historical feature information of patients to predict outcomes. We use 2000 bidirectional LSTM layers to form deep feature extraction network, which can capture features that are more relevant to outcomes.

3) We adopt the peephole connection (shown the red box in Appendix B) to add the long-term state $ct-1$ of the previous moment to the inputs controlled by the input and forget gates, and the long-term state ct of the current moment to the inputs controlled by the output gate, which can better capture historical features related with outcomes of AKI inpatients.

4) In the optimization process of AKIEPM, we adopt the strategies of dropout and L2 regularization, which can avoid overfitting and make the proposed method have good generalization.

We use routine hospital data in the model, for example, demographic, laboratory indicators, diagnostic information, which are consistent in different times of current population, then the proposed method can adapt current population and perform by above reasons.

Meanwhile, we also estimate the performance of AKIEPM for deployment to a certain extent. According to the comments and suggestions of the reviewer, it would be wise to use data most representative of the intended future deployment data. We extract the data from the external validation cohort in 2019-2021, which is close to the current time, and conduct the validation experiment, whose results are shown in the following tables for predicting death and dialysis, respectively.

Death	Accuracy	AUROC	Precision	Recall	F-score
24h	0.9182	0.9158	0.7449	0.9184	0.8226
48h	0.9264	0.9263	0.7333	0.9266	0.8187
72h	0.8038	0.9078	0.8272	0.8038	0.8153
7d	0.8663	0.8721	0.7447	0.8664	0.8009

Dialysis	Accuracy	AUROC	Precision	Recall	F-score
24h	0.8573	0.7262	0.3893	0.9090	0.5451
48h	0.8277	0.7630	0.6527	0.8480	0.7377
72h	0.8652	0.7556	0.4205	0.9196	0.5771
7d	0.8481	0.6948	0.3858	0.9177	0.5432

The results of the data from the external validation cohort in 2019-2021 are close to the results in the total external validation cohort, for example, the AUROC for predicting death within 24h and 48h is 91.58% and 92.63%, respectively, which verify the proposed method also obtains better performance.

Of course, predictive models built on previous data must be clinically validated in prospective studies. Our plan is to conduct a prospective controlled study to assess and independently validate whether our model can improve adverse outcomes in AKI patients. This work is currently being prepared, and we will release the research findings once the research is completed.

Changes made: Page 18, Line 415-426 and Page 19, Line 433-440

4. In addition, you continue to imply that your model will improve outcomes there is no evidence of this. The clinical trial of this algorithm has not been done.

Author response: We appreciate your valuable comments and suggestions. We have revised the statement of limitation and conclusion based on your comments.

Our plan is to conduct a prospective controlled trial to assess and independently validate whether our model can improve adverse outcomes in AKI patients. We also plan to develop a software platform to conduct the clinical trial of this algorithm in future work (Preliminary function and architecture please refer to Appendix D in this document). These works are currently being prepared, and we will release the research findings once the research is completed.

Changes made:

- Line 288-289, under last paragraph of Discussion, “Those who were at risk for AKI could be identified through AKIEPM.”
- Line 277-279, under Discussion, “AKIEPM was conducted on previous data, and whether it can help improve prognosis must be confirmed in prospective controlled study.”

Reviewer #2

1. Authors have answered all my questions satisfactorily. I have no further comments.

Author response: Thank you for your constructive comments and suggestions. Thank you for your affirmation and recommendation for our manuscript.

Supplemental information

Appendix A

The strategies of splitting longitudinal data temporally.

For training the model on the multiply derivation data of different years, we can design an integrated strategy to train the model on different data of different years to avoid overfitting, this is, the model is trained separately on each time data, and then synthesized through the integrated model, such as bagging algorithm, to weighted vote for choosing the results with maximum weight as the predicting results (The schematic figure of the strategy is shown in the following figure).

For testing and validating the model on the multiply data of different years, we can adopt the over-sampling and under-sampling strategies to make the scale of multiply testing and validating data equal. Taking test set as an example, by calculating the average number of multiple sets, if the number of a set is less than the average, we can use over-sampling to increase scale to average. If the number of a set is greater than the average, we can use under-sampling to decrease scale to average. Then we can test and validate the model on them and evaluate its performance on different time data to choose the best model. (The schematic figure of the strategy is shown in the following figure). Here is another strategy, this is, we can design a comprehensive multiple objective function to evaluate the common effect of the scale and result at the same time.

Appendix B

The details of the proposed method AKIEPM.

AKIEPM includes embedding, feature extraction and output layers, whose framework are shown in Figure S1.

Figure S1 Framework of the AKI event prediction model AKIEPM.

(1) Embedding layer

The clinical variables include time, category, and value. Let $t_t^p = \{s_1, x_t^1, \dots, s_i, x_t^i, \dots, s_V, x_t^V\}$, where $s_i \in \mathbb{R}^C$ and $x_t^i \in \mathbb{R}^V$ are the category and value of the i -th variable at time t , respectively. To help AKIEPM capture the dependence across different variables, we embedded the values of clinical variables into continuous vector $E(t_t^p)^{23}$.

(2) Feature extraction layer

This layer is composed of a bidirectional LSTM layer and a one-way LSTM layer.

The first layer employs 2000 bidirectional LSTM layers to extract the features from the forward and reverse of input sequence $x_t = E(t_t^p)$. The bidirectional LSTM is used to extract the hidden features for the embedding of sequential representation based on input, forget, and output units i_t, f_t, o_t of LSTM at time t with

$$i_t = f(W_{ie}x_t + W_{ih}h_{t-1} + w_{ic} \circ c_{t-1} + b_i),$$

$$f_t = f(W_{fe}x_t + W_{fh}h_{t-1} + w_{fc} \circ c_{t-1} + b_f),$$

$$c_t = f_t \circ c_{t-1} + i_t \circ g(W_{ce}x_t + W_{ch}h_{t-1} + b_c),$$

$$o_t = f(W_{oe}x_t + W_{oh}h_{t-1} + w_{oc} \circ c_t + b_o),$$

$$h_t = o_t \circ g(c_t).$$

Here, c_t is cell state, x_t and h_t are the input and hidden representations, respectively. $f()$ adopts sigmoid function and $g()$ adopts tanh nonlinear function with weight $W_{ie}, W_{fe}, W_{ce}, W_{oe}, W_{ih}, W_{fh}, W_{ch}$ and W_{oh} connecting different inputs and gates with memory cells and outputs, as well as biases b_i, b_f, b_c and b_o . Cell state c_t is updated with a fraction of the previous cell state c_{t-1} controlled by f_t , and a new input state is created from the element-wise product \circ of i_t and the output of the cell state by $g()$ to remember the related features of death risk of AKI inpatients from historical records and current state. The peephole connection weights w_{ic}, w_{fc} and w_{oc} further influence the input, forget, and output units.

The next one-way LSTM layer fuses the bidirectional output and obtains the vector representation h_t of all variables with attention mechanism at each moment in feature space, then the hidden representation of t_t^p can be defined as

$$H_t = [h_1, \dots, h_t, \dots, h_T].$$

(3) Output layer

Hidden state H is fed through the *softmax* layer to predict the events in 24h, 48h, 72h, and 7d, respectively.

$$\hat{y}_p^t = \text{softmax}(W_s H_t + b_s).$$

Objective function

Let θ be the set of parameters, prediction probability vector \hat{y}_p^t can be denoted by model posterior distribution $p = (y_p | X(p); \theta)$. We used the cross-entropy between ground truth y_p and prediction probabilities \hat{y}_p^t to calculate the loss as follows,

$$L(\theta) = \frac{1}{N} \sum_{p=1}^N \left((y_p)^T \log(\hat{y}_p^t) + (1 - y_p)^T \log(1 - \hat{y}_p^t) \right).$$

Appendix C

The process of this model step by step with an example.

1) Construction of patient-oriented data. As shown in Supplemental Manuscript, there are standard tables in CRDS, which is recorded on a project-oriented basis, for example, a temperature table records the values of all patients. To analyze the state of a patient at different times, we first construct the patient-oriented data (the patient A with an example shown in the following table), that is, recording all information of a person at one time as an item and organizing these items sequentially according to time point (Since there are many concrete subitems of each type, it is presented by category here. Subitem data of each class can be missing).

Sequential Representation	Time	Day	Demographic	Visit	Vital Signs	Surgery	Laboratory	Prescription	Diagnosis	Event
t_1^A	T1	24h	Yes	Yes	Yes					Yes
t_2^A	T2		Yes	Yes		Yes				Yes
t_3^A	T3		Yes	Yes			Yes			Yes
t_4^A	T4	24h	Yes	Yes				Yes		Yes
t_5^A	T5		Yes	Yes					Yes	Yes

2) Merging data. To decrease the amount of missing data in patient-oriented data, we merged the data in 24h by the merge rules to construct a sequential representation (an example shown in the following table) for dynamically predicting death or dialysis.

Sequential Representation	Time	Demographic	Visit	Vital Signs	Surgery	Laboratory	Prescription	Diagnosis	Event
t_3^A	T3	Yes	Yes	Yes	Yes	Yes			Yes
t_5^A	T5	Yes	Yes				Yes	Yes	Yes

3) Prediction model. The sequential representation t_3^A and t_5^A of patient A are inputted to the AKIEMP through embedding, feature extraction and output, then we can obtain the prediction results.

Step 1. Embedding. The sequential representation t_3^A and t_5^A is embedded to $x_1 = E(t_3^A)$ and $x_2 = E(t_5^A)$.

Step 2. Feature extraction. The bidirectional LSTM and one-way LSTM is used to extract the hidden features h_1 and h_2 for x_1 and x_2 based on input, forget, and output units i_t, f_t, o_t . Then we can the hidden feature of current patient $H_2 = [h_1, h_2]$.

Step 3. Output. Hidden state H_2 is fed through the *softmax* layer to predict the death or dialysis in 24h, 48h, 72h, and 7d, respectively.

4) Evaluation. We use Precision, Recall, F-score, AUROC, and Accuracy to evaluate the model in the derivation, internal validation, and external validation cohorts. Reference Tomašev et al. [1], we adopted the Precision, Recall and AUROC to evaluate the performance of AKIEPM. Meanwhile, we added F-score, which is the comprehensive measurement of Precision and Recall. In addition, we used Accuracy to assist evaluate the performance.

[1] Tomašev et al. Use of deep learning to develop continuous-risk models for adverse event prediction from electronic health records [J]. Nature Protocols, 16: 2765-2787, 2021.

Appendix D

The preliminary plan of software platform to conduct the clinical trial of this algorithm in future work.

①system function

The software platform has three function modules, user management, data management and outcome prediction, as shown in the figure of system function.

(1) user management: implementing basic user information operations, like registration, login and create, read, update, delete user information.

(2) data management: For the clinical deployment, the data of new patients have two way to input the platform, one is manually entering the patient's clinical variable data (data input), which is suitable for small samples, one is automatic importing all clinical variable data with Excel (data import). We also want to directly connect to hospital information system to obtain available data. Meanwhile, the platform can export the prediction results (result export).

(3) outcome prediction: with the input data, the AKIEPM can predict the outcomes (death and dialysis) of AKI inpatients within 24h, 48h, 72h and 7d at each day (death prediction and dialysis prediction). The prediction results can be used to assist clinical intervention, then the intervention results can be feedbacked into the platform to modify prediction model (evaluative feedback).

② system architecture

The software platform adopts Spring + SpringMVC + MyBatis architecture, which separates the input, processing, and output of system to reduce coupling and increase maintainability, as shown in the following figure.

The main logic is

(1) The user input to the software platform,

- (2) the web layer (Controller) takes the input and invokes domain model layer (Model) and user interface layer (View) to fulfill user's requirements,
- (3) the domain model layer conduct AKIEPM and feedback prediction results to user by user interface layer,
- (4) then user decides what action to do next (closed loop),
- (5) The data are stored in database (MyBatis).